# Telling the same story: Fishers and landing data reveal changes in fisheries on the Southeastern Brazilian Coast

**Carine O. Fogliarini**[1]*, **Carlos E. L. Ferreira**[2], **Jéssica Bornholdt**[1], **Moysés. C. Barbosa**[2], **Vinicius J. Giglio**[3], **Mariana G. Bender**[1]

**1** Departamento de Ecologia e Evolução, Marine Macroecology and Conservation Lab, Universidade Federal de Santa Maria, Bairro Camobi, Santa Maria, Rio Grande do Sul, Brazil, **2** Departamento de Biologia Marinha, Reef Systems Ecology and Conservation Lab, Universidade Federal Fluminense, Niterói, RJ, Brazil, **3** Marine Conservation and Ecology Lab, Instituto do Mar, Universidade Federal de São Paulo, Santos, São Paulo, Brazil

* carine_fogliarini@hotmail.com

**Data Availability Statement:** All relevant data are available in GitHub repository (link https://github.com/CarineFogliarini/Fogliarini-et-al-2021).

## Abstract

An understanding of the effects of fishing on marine ecosystems relies on information about the conserved state of these environments. Non-conventional approaches such as the use of historical data and local ecological knowledge can provide information and help adjust our references of changes in the environment. Also, the combination of different types of data can indicate a fisheries trend that would be undetectable when evaluated separately. Here we investigated changes in fisher's perceptions regarding overexploited and new target species in artisanal fisheries in a secular fishing village of the subtropical, southeastern Brazilian coast. We identified temporal changes in landings and in the mean trophic level (MTL) of high trophic level species ($\geq 3.5$ and $>4$) over 16 years. Fishers' knowledge revealed shifts in perception associated with years of fishing practice. More experienced fishers recognized a greater number of overexploited and new target species than fishers in the beginning of their careers. Landing data has revealed declining trends of 72% for five mesopredators species. Due to the overfishing of mesopredators, there was a shift in target species, towards fish that were previously discarded. Temporal changes in landings and in the MTL metric are concordant with previous reports on the overexploitation of species caught by local fishers. Our work reveals that multiple sources of information can be combined to establish historical baselines and improve the detection of change in marine ecosystems.

## Introduction

Marine and coastal ecosystems are globally threatened due to human impacts such as overfishing, pollution, habitat destruction and climate change [1, 2]. Indeed, anthropogenic activities have been changing the oceans for millennia, with fishing being the most ancient activity performed by humans [3]. Some types of fishing have been unsustainable [4], surpassing the recovery limits of several fish stocks across the globe [5]. The status assessment of fishery

**Funding:** C.O.F. acknowledges Coordenação de Aperfeiçoamento de Pessoal de Nível Superior (CAPES) for financial support. V.J.G. received a post-doctoral grant #2017/22273-0, São Paulo Research Foundation (FAPESP). CELF is supported by grants from FAPERJ and CNPq. Costão Rochoso Project (FUNBIO Grant Pesquisa Marinha 020/2017) has provided financial, and logistic support during field sampling in Arraial do Cabo. This work is part of a research program supported by Serrapilheira Institute (grant number Serra-1708-15364) awarded to Guilherme Ortigara Longo (PI).

**Competing interests:** NO authors have competing interests

resources revealed that the fraction of fish stocks that are within biologically sustainable levels decreased from 90%, in 1974, to 65.8% in 2017 [6]. In contrast, recent studies argue that some fishing stocks are gradually recovering in different parts of the world [7], especially in regions where there are effective fisheries management [8]. Although these recent findings show positive global scenarios [9], overfishing is responsible for substantially modifying the abundance and biomass patterns of species and decreasing the body size of individuals in a population [10–13]. Some fishing gears such as longline are usually directed towards top predators (e.g. tunas and sharks) and select individuals with larger body sizes [14], a common pattern with known consequences for marine ecosystem functioning, including the prey release effect [15, 16].

Fishing effort concentrated on large-bodied fish can cause the decline of top predators stocks such as groupers, sharks and tunas [17–21], and even drive species to local extinction [3, 22]. These large-bodied species tend to occupy higher levels in the food chain and play a key role in maintaining the trophic structure and functioning of communities [23, 24]. Thus, the removal of top predators can trigger cascading effects on lower levels of the food web [25, 26]. Declining trends of higher trophic level stocks lead to a systematic replacement by lower trophic level and smaller-bodied target species, a process identified as "fishing down the food web" [27–29]. This process has been measured through a decline in fisheries' mean trophic level (MTL) [30–33]. In an attempt to understand how fishing impacts marine food webs, studies highlight that there are other ways in which fishing may be affecting the ecosystem. In "fishing through the food web" a decline in MTL is caused due to the sequential addition of low trophic level rather than to a decline of high trophic level catches [25]. The "fishing up the food web" process corresponds to a shift from low to high trophic level species over time, causing an increase in the MTL of catches [34, 35]. Multiple hypotheses may explain how fishing affects marine food webs [36], such as profit-driven fishing [37]. The use of catch MTL as an indicator to measure the magnitude of fishing impacts on marine ecosystems may not be reliable in some scenarios, and should be interpreted cautiously [38]. The use MTL in combination with other measures may offer a better picture of the ecosystem [39].

The assessment of temporal changes in MTL and replacement of target species requires long-term fisheries monitoring. However, most low and lower-middle income countries lack fisheries monitoring programs [40]. In Brazil, such programs are rare and fragmented, and since 2008, there is no governmental fisheries monitoring program established at the national level. In addition, lack of financial resources, a large (~8000 km) and heterogeneous coast, a multigear fishing scenario, differential efforts and target species, pose a challenge to the implementation of fisheries management strategies in Brazil [41, 42]. To circumvent the difficulties caused by the lack of fishing data, non-conventional approaches can be useful to estimate the impacts of fishing over time [43–45]. For instance, local ecological knowledge (LEK) may provide a relevant source of historical information, in which marine resource users (e.g. fishers) experienced different scenarios in the past [44, 46, 47].

A robust assessment of the magnitude and effect of anthropogenic impacts (e.g. fishing) on ecosystems requires knowledge of the unexplored state of these environments [48]. However, throughout human generations, information on composition, size and abundance of species caught on fisheries may be lost, compromising the environmental perception adopted by human populations [49]. Such cross-generational change in baselines has implications to environmental issues, as it results in increased tolerance for progressive environmental degradation because the younger generations consider a more degraded environment as the norm [50, 51]. Pauly et al. [50] described this phenomenon as the "shifting baseline syndrome", a socio-psychological tendency to assume that "natural" conditions of an ecosystem are those first observed by users, researchers and contemporary managers. When data on the conservation

status of the environment is scarce, historical data from alternative sources (e.g. works of art, reports of naturalists, logbooks, photographs, grey literature, and anecdotal information) have been used to help adjust references [52].

Fisher's LEK has been widely used to investigate the past of fisheries, mainly in data poor countries. The empirical knowledge of natural resource users is a cost-effective alternative approach that offers relevant data to inform management. For instance, fishers LEK has provided information on temporal and spatial patterns of catches [53–55], behavior of targeted species [42, 56–59], changes in abundance and body size of targeted species [20, 60], trophic and ecosystem models [61, 62], and conflicts with other resource users and management interventions [63, 64]. In certain cases, LEK has been also associated with other types of data, such as landing data and underwater visual census, to reveal temporal trends in fish populations [54, 61, 65–67]. These combined approaches offer an opportunity to understand the effect of small-scale fisheries in data poor scenarios–when temporal and systematically collected data are missing. Combining anecdotal evidence from fisher's LEK and fisheries landing data, we aim to identify temporal changes in (i) small-scale fisher's perception regarding overexploited and new target fish species; (ii) and landings of overexploited and new target fish species. Specifically, we combined the MTL metric with fishers' perceptions to reveal temporal changes in local trophic structure caused by fishing. We also discuss the main causes of fish overexploitation and management strategies in a subtropical fisheries hotspot.

## Materials and methods

### Study site

The study was conducted in Arraial do Cabo, southeastern Brazil (Fig 1). Arraial do Cabo is an ancient Brazilian fishing area, where a variety of artisanal fishing gear is employed such as hook and line, gillnet, spearfishing and beach seine. In Arraial do Cabo, artisanal fisheries still

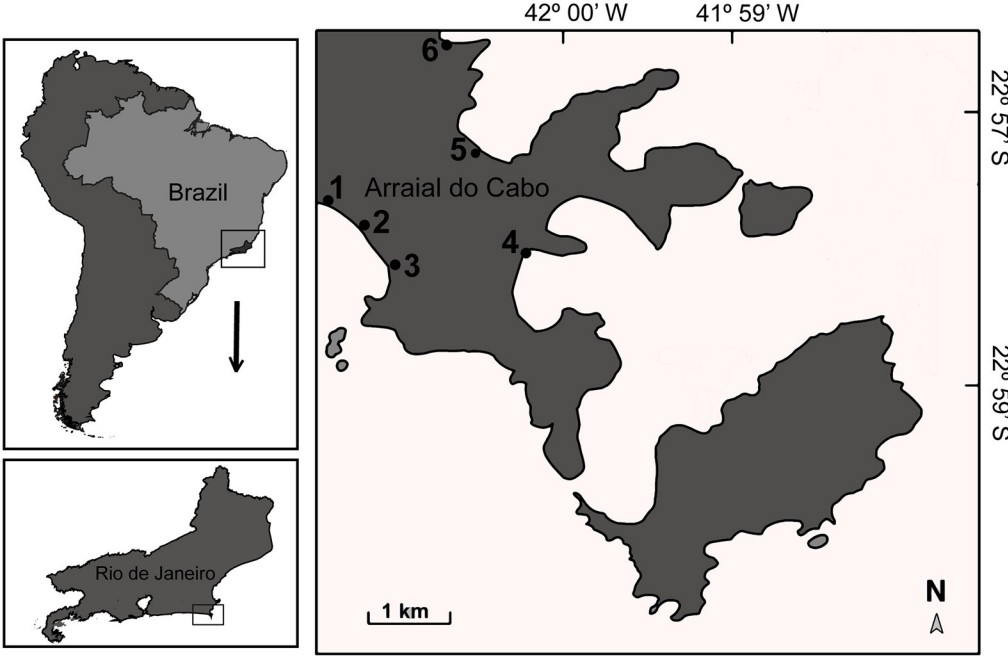

**Fig 1. Map of Arraial do Cabo region (state of Rio de Janeiro, Brazil), showing interview sites.** Map reprinted from [54] under a CC BY license, with permission from Bender et al (2014), original copyright 2014.

occur with small boats such as canoes some without any motors. Fishers make trips near the shore, or using hook and line or small nets like cast nets in the rocky shore or on the beach [68]. The beach seine is an old/secular practice that was central to the establishment of the local partially protected marine protected area (MPA)–the Arraial do Cabo Marine Extractive Reserve. In this MPA, only local fishers are allowed to fish, with 1600 families relying on fishing activities as the main source of income [68]. Arraial do Cabo holds a diverse marine community, gathering tropical and warm temperate components, which is favored by the coastal morphology and local upwelling events [69, 70].

## Data collection

**Fisher's local ecological knowledge.** Between July and August 2018 and 2019, we interviewed 155 artisanal fishers on the fishing communities of Figueira, Monte Alto, Praia Grande, Praia dos Anjos, Prainha, and Pontal (Fig 1). Interviewed fishers correspond to 10.3% of local artisanal fishers (n = 1500) [71]. Interviews were conducted following previous consent and individually, so that there was no influence from others on fishers' answers. Semi-structured questionnaires included questions on (i) which fish species were previously discarded but are now fishing targets; (ii) which species are overexploited in the region; and (iii) what are the main causes of species' overexploitation (S1 File in S1 Data). The fishers answered spontaneously which species they consider overexploited and new targets of fisheries. Fishers were randomly approached at Praia dos Anjos pier, and invited to participate in the interviews. This procedure was conducted until reaching an appropriate number of interviews (n). When completing the questionnaire (S1 File in S1 Data), fishers were asked to indicate another interviewee [72].

We also asked fishers their age and years of fishing experience. For data analysis, we chose to use the fisher's experience (time period) rather than age, as these two variables are correlated ($r^2$ = 0.53, cor = 0.74, $P < 0.001$, Pearson's correlation test). Fishers were categorized according to their fishing experience into four groups: less experienced (<20 years, n = 24), intermediate (21–35 years, n = 55), experienced (36–40 years, n = 44) and very experienced (>40 years, n = 32). Under this classification, we consider only their time of experience (years of practice), with no qualitative judgment of the real experience of each fisher [73]. We assume that within a given category of experience, all fishers spend the same time fishing, and all gain experience at the same rate [73]. Still, this set of different categories can help estimate changes in fish composition as well as differences in perceptions of trends across generations of fishers. In addition, we chose these experience categories so that they would accommodate similar sample sizes. The small sample size of the less experienced category compared to the very experienced is related to the lack of engagement with fishing by the younger generations. Younger fishers are switching to alternative sources of income such as tourism, often encouraged by their families to leave the fishing [71]. The interviews were approved by the Ethics Committee of Universidade Federal de Santa Maria, Brazil (CAAAE 29157919.6.0000.5346) and by Sistema de Autorização e Informação em Biodiversidade (Sisbio-ICMBio/IBAMA/Brazil nr. 55911–6).

**Fisheries landing data.** We searched for landing data on the main species reported by the interviewees. Landing data (e.g. in kg) has been incorporated into the study to complement the trends identified by local fishers. Such data corresponds to local annual landings of artisanal fisheries registered between 1992 and 2008 in the port of Arraial do Cabo (Fisheries landing monitoring report, FIPAC).

**Mean trophic level.** We compiled the trophic level (TL) of 81 species landed in Arraial do Cabo between 1992 and 2008 (S1 Table in S1 Data). The trophic levels were obtained from FishBase, which provides TL estimates from food items for many fish species [74]. For the TL of invertebrates, we used the equivalent database, SeaLifeBase [75].

To assess MTL trends landings over time, we compared the MTL of species in five groups: all landed species, species with TL > 4, TL ≥ 3.5, and TL < 3.5. These categories allow us to assess changes in MTL taking into account the temporal dynamics of higher and lower trophic level species, and visualize potential masking effects [31]. MTL was also calculated excluding low TL species (TL < 3.5), because of possible fisheries expansion in direction to small pelagic species, which can represent an alternative scenario such as 'fishing through' food webs rather than 'fishing down' [25]. High trophic level species (TL > 4) are more sensitive to overfishing, as they have life history traits such as late maturation, low fertility rates, and slow growth [76]. Therefore, we examined the dynamics of species' stocks with a threshold value of TL > 4, as done previously [25]. Species with TL > 3.5 (e.g. tunas, cod and groupers) are considered as mesopredators that feed on a mixture of low and high-TL organisms [77]. Also, species with TL < 3.5 were assessed to observe changes in the contribution of low trophic level species [77]. We used these threshold values to eliminate the masking effect and to emphasize changes in the relative abundance of fish from certain trophic groups. The MTL was estimated as follows [27]:

$$MTL = \frac{\sum TL_{ij} Y_{ij}}{\sum Y_{ij}}$$

Where MTL is the mean trophic level of landing in year $j$, $Y_{ij}$ the landing of species $i$ in year $j$ and $TL_i$ is the trophic level of species $i$.

We used a null model approach to test whether citations of overexploited and new target species (by fishers), within a given trophic level, are more frequent than expected at random. We generated 1000 random samples for the number of citations of each species within the overexploited and new target species groups. For each random sample, we calculated the MTL weighted by the citation number, using the following equation:

$$MTLc = \frac{\sum TL_i\, C_i}{\sum C_i}$$

Where $MTL_c$ is the mean trophic level of overexploited and new target species and $C_i$ is the total number of citations of species $i$.

We then plotted the confidence intervals (C.I. 95%) of the null MTL obtained for overexploited and new target species and the observed $MTL_c$, to verify if the citation frequency differs than expected at random.

## Data analysis

To estimate changes in fishers' perception of target species, we applied a Generalized Linear Model (GLM) between the (i) number of overexploited species and (ii) number of new target species vs. experience of fishers (years of practice). The number of cited species was set as the response variable. Because numbers of species are count data, we assumed a Poisson distribution. The experience represents the explanatory variable. Under these specifications, the equation of model was:

$$log(\mu i) = \beta 0 + \beta 1\ xi$$

Where $\mu i$ is the conditional expectation of $y_i$, E(y|x), $\beta 0$ is the coefficient marked (Intercept) and $\beta 1$ the coefficient marked $x$. We assumed the model of the mean number of species $\mu i$ using a log link.

We measured the mean percent of increase in citations (P) of overexploited fish species and new target species according to fishers' experience (F) using the GLM model coefficients

(*coef1* and *coef2*):

$$P = \left( \frac{e^{(coef1+coef2*F)}}{e^{(coef1+coef2*0)}} - 1 \right) * 100$$

To test for differences between the proportion of overexploited and new target species (previously discarded) mentioned by fishers in different experience categories, we applied a Z-test. We then verified differences between groups through a pairwise comparison of proportions. Regression analyses were used to measure temporal trends in landings of fish species by plotting the landed weight (in tons) over the years. All analyzes were performed on R software v. 3.6.1 [78].

## Results

Fishers with more years of practice recognized a significantly greater number of overexploited fish species than those with fewer years of practice ($\beta$ = 0.009, df = 153, *P* = 0.008) (Fig 2A, S2 Table in S1 Data). This relationship is clearly detected by an increase in the mean percent in the citation, related with years of practice (Fig 2B). We observed the same pattern for the number of species recognized as new target species. As time of practice increases, fishers mentioned a significantly greater number of species as new targets of local fisheries ($\beta$ = 0.010, df = 153, *P* = 0.01) (Fig 2C, S2 Table in S1 Data). In addition, there is an increase in the percent of citations of new target species by fishers' with more years of practice (Fig 2D).

In total, 37 species were identified as overexploited by local fishers (Table 1). The bluefish (*Pomatomus saltatrix*) was the most frequently cited species, across all experience categories. The proportion of citations of *P. saltatrix* were similar between categories, with at least 45% of fishers in different experience categories recognizing bluefish as overexploited: 56% of very experienced (n = 18), 55% of experienced (n = 21), 47% of intermediate (n = 18), and 45% of less experienced fishers (n = 21) (Fig 3A). The second most cited species was the largehead hairtail (*Trichiurus lepturus*), although there are no significant differences between the experience categories, the species was mentioned more by intermediates (26%, n = 10) and less experienced (30%, n = 14) fishers. The dusky grouper (*Epinephelus marginatus*) and the crevalle jack (*Caranx hippos*) were most frequently cited by very experienced fishers (> 40 years of practice) and had a significant difference in the proportion of citations between very experienced and beginners (*P* = 0.03 and *P* = 0.001, respectively). In addition to *C. hippos*, the lesser amberjack (*Seriola fasciata*) was another carangid referred to as overexploited (Fig 3A).

Fishers identified 36 new target species that were previously discarded by local fisheries (Table 2). *T. lepturus* was cited by 63% of very experienced fishers, 76% of experienced, 71% of intermediates, and 66% of less experienced fishers (Fig 3B). The second most cited species was the grey triggerfish (*Balistes capriscus*), followed by the Argentine conger (*Conger orbignyanus*), the unicorn leatherjacket filefish (*Aluterus monoceros*) and the Atlantic bigeye (*Priacanthus arenatus*) (Fig 3B). For *T. lepturus* and *C. orbignyanus* the proportion of citations was similar across experience categories (Fig 3B). The citations for *P. arenatus* decreased with fisher's experience, therefore being greater for less experienced fishers. Only the citations of *B. capriscus* had significant differences between experience categories, being proportionally higher for experienced compared to intermediate fishers (*P* = 0.01) (Fig 3B).

The same declining trend reported by fishers for *P. saltatrix*, *T. lepturus*, *E. marginatus*, *C. hippos*, and *S. fasciata* (Fig 3A) were revealed by fisheries landing data (Fig 4). Since the 1990s, *P. saltatrix* stocks declined significantly (i.e. approximately 47%) (Fig 4A), this status being perceived by fishers across all experience categories. For *T. lepturus*, the landing declined significantly after the 2000s (Fig 4B), pattern promptly identified by the majority of fishers in less

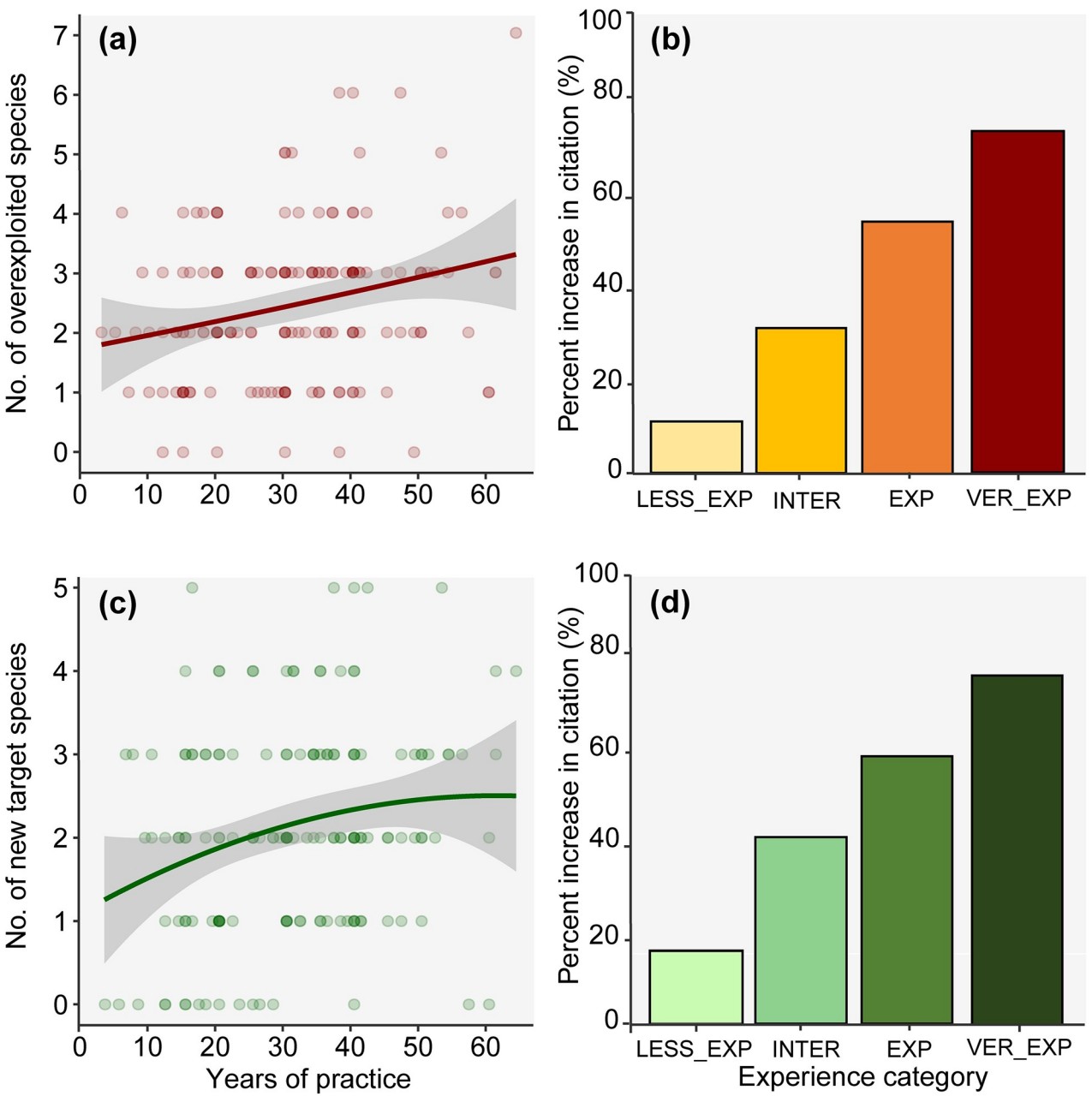

**Fig 2.** (a) Number of species mentioned as overexploited according to fisher's experience (years of practice); (b) Probability of citations of overexploited species according to fisher's experience; (c) Number of species mentioned as new target species according to fisher's experience; and (d) Probability of citations of new target species according to fisher's experience. Experience categories: LESS_EXP = less experienced; INTER = intermediate; EXP = experienced; and VER_EXP = very experienced.

experienced categories. This species has been considered overexploited and also identified as a new target species in the region. Landing data show that since the 1990s, *E. marginatus* declined 61%, *C. hippos* diminished by 88%, and *S. fasciata* by 89%, the latter showed significant declining trends (Fig 4C, 4E and 4G). This overexploitation scenario was observed by very experienced fishers, which cited the first two species as overexploited in a higher proportion (Fig 3A).

**Table 1. Species reported as overexploited by fishers in descending order of number of citations (N).**

| | OVEREXPLOITED SPECIES | | | | | | |
|---|---|---|---|---|---|---|---|
| Order | Species | N | % | Trophic level | Trophic category | Global status | Regional status |
| 1 | *Pomatomus saltatrix* | 78 | 50.3 | 4.5 | MCAR | VU | NT |
| 2 | *Trichiurus lepturus* | 37 | 23.8 | 4.4 | MCAR | LC | LC |
| 3 | *Epinephelus marginatus* | 23 | 14.8 | 4.4 | MCAR | VU | VU |
| 4 | *Seriola fasciata* | 22 | 14.1 | 4.5 | MCAR | LC | DD |
| 5 | *Caranx hippos* | 20 | 12.9 | 3.6 | MCAR | LC | LC |
| 6 | *Pseudopercis numida* | 20 | 12.9 | 3.9 | MCAR | LC | NT |
| 7 | *Hyporthodus niveatus* | 16 | 10.3 | 4 | MCAR | VU | VU |
| 8 | *Seriola lalandi* | 14 | 9 | 4.2 | MCAR | LC | LC |
| 9 | *Mugil liza* | 11 | 7 | 2 | HERB | DD | NT |
| 10 | *Scomberomorus cavalla* | 11 | 7 | 4.4 | MCAR | LC | LC |
| 11 | *Sardinella brasiliensis* | 10 | 6.4 | 3.1 | PLANK | DD | DD |
| 12 | *Conger orbignyanus* | 10 | 6.4 | 3.7 | MCAR | LC | DD |
| 13 | *Sarda sarda* | 8 | 5 | 4.5 | MCAR | LC | LC |
| 14 | *Cynoscion striatus* | 7 | 5 | 3.9 | MCAR | NE | NE |
| 15 | *Mycteroperca microlepis* | 6 | 3.8 | 3.7 | MCAR | VU | DD |
| 16 | *Coryphaena hippurus* | 6 | 3.8 | 4.4 | MCAR | LC | LC |
| 17 | *Seriola dumerili* | 6 | 3.8 | 4.5 | MCAR | LC | LC |
| 18 | *Priacanthus arenatus* | 5 | 3.2 | 4 | MINV | LC | LC |
| 19 | *Katsuwonus pelamis* | 5 | 3.2 | 4.4 | MCAR | LC | LC |
| 20 | *Caranx crysos* | 4 | 2.5 | 4.1 | MCAR | LC | LC |
| 21 | *Caranx latus* | 4 | 2.5 | 4.2 | MCAR | LC | LC |
| 22 | *Pagrus pagrus* | 3 | 1.9 | 3.9 | MINV | LC | DD |
| 23 | *Trachinotus carolinus* | 3 | 1.9 | 3.5 | MINV | LC | LC |
| 24 | *Micropongonias furnieri* | 3 | 1.9 | 3.1 | MINV | LC | LC |
| 25 | *Cynoscion acoupa* | 3 | 1.9 | 4.1 | MCAR | LC | NT |
| 26 | *Lopholatilus villarii* | 2 | 1.2 | 3.8 | MCAR | NE | VU |
| 27 | *Balistes vetula* | 2 | 1.2 | 3.8 | MINV | NT | NT |
| 28 | *Diplodus argenteus* | 2 | 1.2 | 3.1 | OMNI | LC | LC |
| 29 | *Peprilus paru* | 2 | 1.2 | 4.5 | MCAR | LC | LC |
| 30 | *Centropomus undecimalis* | 2 | 1.2 | 4.2 | MCAR | LC | LC |
| 31 | *Scomber colias* | 1 | 0.6 | 3.4 | MCAR | LC | LC |
| 32 | *Scomberomorus brasiliensis* | 1 | 0.6 | 3.3 | MCAR | LC | LC |
| 33 | *Aluterus monoceros* | 1 | 0.6 | 3.8 | OMNI | LC | NT |
| 34 | *Lagocephalus laevigatus* | 1 | 0.6 | 4 | MCAR | LC | LC |
| 35 | *Carcharhinus leucas* | 1 | 0.6 | 4.3 | MCAR | NT | NT |
| 36 | *Isurus oxyrinchus* | 1 | 0.6 | 4.5 | MCAR | NT | NT |
| 37 | *Pseudobatos horkelii* | 1 | 0.6 | 3.8 | MINV | CR | CR |

Percentage of citation (%) of species by fishers (n = 155). Trophic levels were obtained from FishBase where a higher value means higher trophic levels [79]. Trophic categories were obtained from the Southwestern Atlantic reef fish database [80]. The conservation status at global level derives from the IUCN Red List of Threatened Species [81] and at national level from the Brazilian Red List of Threatened Species [82].

Trophic categories: Macrocarnivore (MCAR); Mobile Invertivore (MINV); Omnivore (OMNI); Planktivore (PLANK); Herbivore (HERB). IUCN categories: Critically endangered (CR); Vulnerable (VU); Near threatened (NT); Least Concern (LC); Data deficient (DD); Not evaluated (NE).

The landings data of the new target species exhibited an increase from the 2000s onwards, followed by a reduction in the quantities of landed fish over the years (Fig 4B, 4D, 4F and 4H). From the 1990s to 2000s, the landings of new target species as *T. lepturus*, *C. orbignyanus*, and

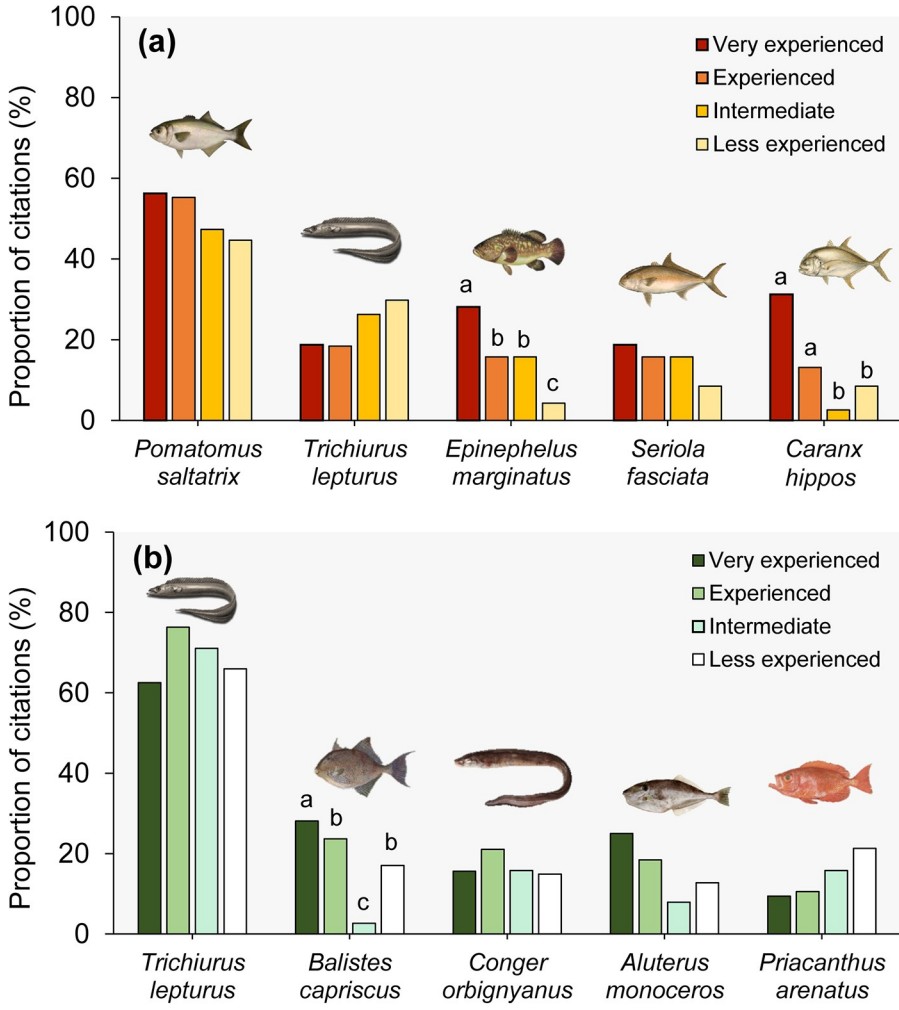

**Fig 3.** Fish species most frequently cited as overexploited (a) and new target species (b) according to fisher's experience (years of practice). Letters indicate significance ($P < 0.05$) between categories of experience of each species.

*A. monoceros* had an increase (Fig 4B, 4D and 4H). By the mid-2000s, landings of these species declined 69%, 55% and 85%, respectively (Fig 4B, 4D and 4H). Fishers also reported this trend, when respondents mentioned these species as overexploited (Fig 3B; Table 1). Landings of *P. arenatus* had a significant increase from 2000 onwards (Fig 4F), mainly reported by less experienced fishers (Fig 3B).

Catch MTL of TL $\geq$ 3.5 and TL $>$ 4 species declined significantly from 1992 to 2008 ($r^2$ = 0.26; $P$ = 0.03 and $r^2$ = 0.43; $P < 0.001$, respectively) (Fig 5A and 5B). When MTL of all landed species are represented, there were non-significant trends in MTL over time ($r^2$ = 0.06; $P$ = 0.32). For species with TL $<$ 3.5, the increasing trend was non-significant ($r^2$ = 0.07; $P$ = 0.14) (Fig 5A and 5B).

In the years of 1995 and 1999, there were peaks in landings of all species in Arraial do Cabo (Fig 5C). However, between 2001 and 2004, there were sharp declines in landings for all groups (Fig 5B). Declining trends were also identified for species with TL $>$ 4 through landing data. For the top 10 species, we observed a decrease in landings between 1992 and 2008 (Fig 5D).

When contrasting the MTL of species mentioned by fishers as overexploited and as new target species, we observed that the citations of overexploited species were more frequently within

**Table 2. Species cited as new targets by fishers, in descending order of reports (N).**

| Order | Species | N | % | Trophic level | Trophic category | Global status | Regional status |
|---|---|---|---|---|---|---|---|
| | | | | NEW TARGET SPECIES | | | |
| 1 | *Trichiurus lepturus* | 107 | 69 | 4.4 | MCAR | LC | LC |
| 2 | *Balistes capriscus* | 27 | 17.4 | 4.1 | MINV | VU | NT |
| 3 | *Conger orbignyanus* | 26 | 16.7 | 3.7 | MCAR | LC | DD |
| 4 | *Aluterus monoceros* | 24 | 15.4 | 3.8 | OMNI | LC | NT |
| 5 | *Priacanthus arenatus* | 23 | 14.8 | 4 | MINV | LC | LC |
| 6 | *Scomber colias* | 13 | 8.3 | 3.4 | MCAR | LC | LC |
| 7 | *Sardinella brasiliensis* | 12 | 7.7 | 3.1 | PLANK | DD | DD |
| 8 | *Katsuwonus pelamis* | 11 | 7 | 4.4 | MCAR | LC | LC |
| 9 | *Lagocephalus laevigatus* | 9 | 5.8 | 4 | MCAR | LC | LC |
| 10 | *Diplodus argenteus* | 8 | 5.1 | 3.1 | OMNI | LC | LC |
| 11 | *Porichthys porosissimus* | 5 | 3.2 | 3.7 | MINV | NE | LC |
| 12 | *Mugil liza* | 4 | 2.5 | 2 | HERB | DD | NT |
| 13 | *Pomatomus saltatrix* | 4 | 2.5 | 4.5 | MCAR | VU | NT |
| 14 | *Micropogonias furnieri* | 4 | 2.5 | 3.1 | MINV | LC | LC |
| 15 | *Cynoscion striatus* | 3 | 1.9 | 3.9 | MCAR | NE | NE |
| 16 | *Umbrina canosai* | 3 | 1.9 | 3.9 | MINV | NE | LC |
| 17 | *Selene setapinnis* | 3 | 1.9 | 3.7 | MCAR | LC | LC |
| 18 | *Dactylopterus volitans* | 3 | 1.9 | 3.7 | MINV | LC | LC |
| 19 | *Trachinotus carolinus* | 2 | 1.2 | 3.5 | MINV | LC | LC |
| 20 | *Caranx latus* | 2 | 1.2 | 4.2 | MCAR | LC | LC |
| 21 | *Cynoscion acoupa* | 2 | 1.2 | 4.1 | MCAR | LC | NT |
| 22 | *Chloroscombrus chrysurus* | 2 | 1.2 | 3.5 | PLANK | LC | LC |
| 23 | *Seriola fasciata* | 2 | 1.2 | 4.5 | MCAR | LC | DD |
| 24 | *Pseudopercis numida* | 2 | 1.2 | 3.9 | MCAR | LC | NT |
| 25 | *Carcharhinus plumbeus* | 1 | 0.6 | 4.5 | MCAR | VU | CR |
| 26 | *Isurus oxyrinchus* | 1 | 0.6 | 4.5 | MCAR | NT | NT |
| 27 | *Squatina argentina* | 1 | 0.6 | 4.1 | MCAR | EN | CR |
| 28 | *Sparisoma axillare* | 1 | 0.6 | 2 | HERB | DD | VU |
| 29 | *Epinephelus marginatus* | 1 | 0.6 | 4.4 | MCAR | VU | VU |
| 30 | *Carcharhinus brevipinna* | 1 | 0.6 | 4.2 | MCAR | VU | DD |
| 31 | *Cynoscion jamaicensis* | 1 | 0.6 | 3.8 | MCAR | LC | LC |
| 32 | *Pseudobatos horkelii* | 1 | 0.6 | 3.8 | MINV | CR | CR |
| 33 | *Lopholatilus villarii* | 1 | 0.6 | 3.8 | MCAR | NE | VU |
| 34 | *Cynoscion guatucupa* | 1 | 0.6 | 3.7 | MCAR | NE | LC |
| 35 | *Pagrus pagrus* | 1 | 0.6 | 3.9 | MINV | LC | DD |
| 36 | *Sarda sarda* | 1 | 0.6 | 4.5 | MCAR | LC | LC |

Percentage of citation (%) of species by fishers (n = 155). Trophic levels were obtained from FishBase where a higher value means higher trophic levels [79]. Trophic categories were obtained from the Southwestern Atlantic reef fish database [80]. The conservation status at global level derives from the IUCN Red List of Threatened Species [81] and at national status from the Brazilian Red List of Threatened Species [82].

Trophic categories: Macrocarnivore (MCAR); Mobile Invertivore (MINV); Omnivore (OMNI); Planktivore (PLANK); Herbivore (HERB). IUCN categories: Critically endangered (CR); Vulnerable (VU); Near threatened (NT); Least concern (LC); Data deficient (DD); Not evaluated (NE).

higher TL species relative to new target ones (S1 Fig in S1 Data). In addition, the MTL observed for overexploited species was higher than expected by chance (S1 Fig in S1 Data).

Fishers mentioned 35 different causes of fisheries overexploitation in Arraial do Cabo (1.58 ± 1.08, mean ± standard deviation). The five most mentioned causes were the increased

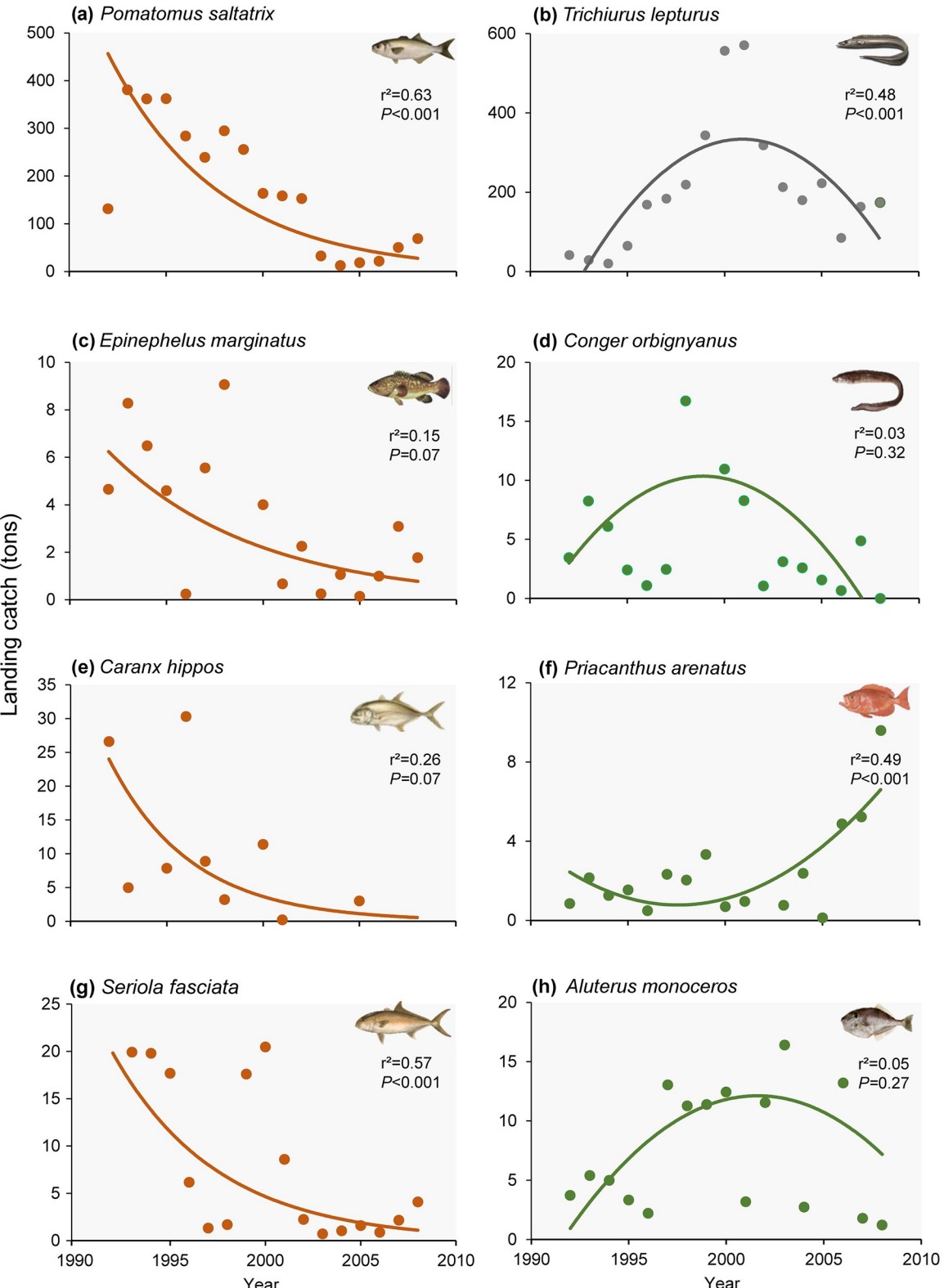

**Fig 4. Artisanal fisheries (tons) landings in Arraial do Cabo from 1992 to 2008.** Species on the left (a, c, e, g) correspond to those most frequently cited as overexploited in interviews and on the right (d, f, h), new target species reported by local fishers. (b) *Trichiurus lepturus* corresponds to both an overexploited and a new target species. a, b, d, f, g and h, are second-order polynomial regressions; c and e, an exponential regression.

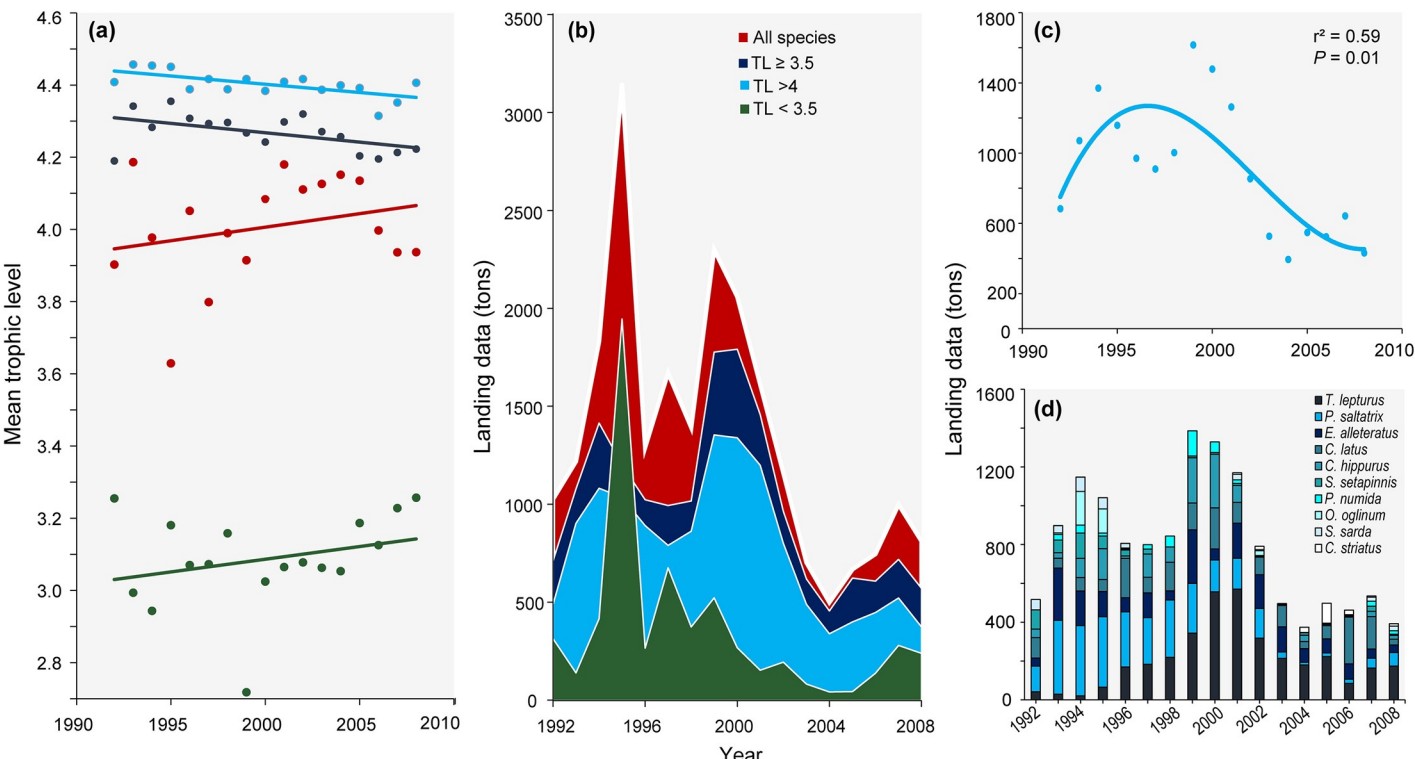

**Fig 5. Trends in MTL of species landed in Arraial do Cabo from 1992 to 2018.** (a) Linear regression of catch MTL; (b) Landing data in tons; (c) Third—order polynomial regression of landings (in tons) of species with TL > 4; and (d) Landings of the top 10 landed species with TL> 4. The colors represent MTL calculated after excluding all species below trophic level 4 (blue), calculated after excluding all species below trophic level 3.5 (yellow), calculated with all species landed (red), and for species with trophic level below 3.5 (green).

fishing effort (22%), which was recognized by fishers as an increase in the number of fishers and boats; presence of industrial fishing vessels (21%), presence of purse seine fisheries (19%), gillnet fishing (16%), and trawling (12%).

## Discussion

Our study revealed a replacement of target species identified by fisher's knowledge and landing data in a Brazilian secular fishing village. Here, we showed an overexploitation scenario of large and valuable fish species and their replacement by new target species. The decline of species stocks such as *P. saltatrix*, *E. marginatus*, *C. hippos* and *S. fasciata* were followed by sequential substitutions by less commercially valuable but more abundant species (e.g., *T. lepturus*, *B. capriscus*, *A. monocerus* and *P. arenatus*). Temporal changes in the landing composition of fish species have been reported around the world [29, 83–85]. In general, changes are from large-bodied species belonging to high trophic levels to smaller and lower trophic level species [10], and/or replacement of fish to invertebrate species [29]. In southern Brazil, a study used fishing landings, market values and multispecies indicators including MTL to evaluate changes and fishing impacts over time [86]. As in our study, authors did not identify declines in MTL of landings. However, they report the decline of most high trophic level species, as sharks, indicated from market data and landing volumes. This study also suggests that changes based on a single indicator such as MTL can be masked, and the use of multiple approaches can make these changes more explicit [86]. In this context, our study is the first to indicate

temporal changes in the composition of fishing resources by combining fisher's knowledge and landing data in Southeastern Brazilian Coast.

Our results show that fisher's perception of overexploited and new target species is shaped by their fishing experience. As experience increases with years of fishing practice, fishers are able to recognize a greater number of overexploited and new target species. These findings indicate the occurrence of a shifted baseline among fishers' generations, which in this case, is the loss in fisher perception regarding changes in fisheries' composition and the decline of fish species throughout generations [50]. At the time young and less experienced fishers started fishing, several species were already overexploited, compromising their current perception of fish stocks. The small number of species reported by less experienced fishers compared with experienced ones may reflect the lack of familiarity and/or a reduced interaction of fishers with the natural environment [49]. Such disconnection with past conditions has effects on the willingness and motivation of new generations to accept management restrictions aimed to restore resources and even mislead recovery perceptions, the "placebo effect" [87]. In Arraial do Cabo, the overexploitation of fishing resources [21, 54] has made less experienced fishers look for alternative income sources. Several less experienced fishers have mentioned being involved in tourism related activities, using their boats for sightseeing, as a parallel activity to fishing. This reduces the engagement of young people with fishing and possibly impairs their perception of the status of fishing resources. The loss of familiarity with the natural environment of future generations can result in the absence of past information or experience with historical conditions [49], and members of each new generation accept the scenario in which they were raised as being normal.

Our results revealed declining trends in landings of commercially important mesopredators in Arraial do Cabo. Declines of high trophic species like groupers and bluefish have been reported over the last decades [54]. *P. saltatrix* and *E. marginatus*, one being pelagic and one a reef-associated, are important resources in Arraial do Cabo, as well as along the south and southeastern Brazilian coast [88]. Our findings highlight that very experienced fishers have identified the overexploited status of *E. marginatus* and *C. hippos* populations in a greater proportion. More experienced fishers have reported higher catches of target species than less experienced ones [17, 20, 53, 60, 73, 89], this being a common scenario of overexploitation in small-scale fisheries since fish resources are decreasing. For instance, decades ago, groupers were abundant in Arraial do Cabo, and generations of more experienced fishers witnessed large catches of the species in the past. Today, older fishers perceive this decline, and catches of groupers have become rare events in the region. Due to life history traits such as high longevity, slow growth, late maturation, and low reproductive rates, groupers are highly sensitive to fishing effects [90, 91]. Such characteristics have contributed to the threatened status of many grouper species worldwide [15, 92]. The decline of other predator species as *C. hippos* and *S. fasciata* is concerning, since these species are important regional fishing resources [93], and as expected for large carnivores, depict a combination of biological traits that can enhance their extinction risk [94].

Large catches of *T. lepturus* in Arraial do Cabo occurred incidentally in the past, i.e. when fishers targeted other species with high commercial value such as *P. saltatrix* and *E. marginatus*. The largehead hairtail was identified both as overexploited and new target species, mostly by less experienced fishers. Fishing of this species is more recent than of other fishes mentioned as overexploited. Thus, knowledge about the exploitation status of this species is more widespread among young generations. Due to its low commercial value and lack of consumer market, large quantities of *T. lepturus* were previously discarded. Later on, the decline of large predators in the region [21, 54], led fishers to replace their target species. This has made *T. lepturus* economically valuable and a target of local fisheries, which was also observed for *B.*

*capriscus*, *C. orbignyanus* and *A. monoceros*. Fishing directed at other less valuable species has become an alternative to maintain local fishing activities. The collapse of higher trophic level species caused by overfishing led fishers to target less valuable and lower trophic level species, similar trends are found in case studies that report the occurrence of fishing down marine food webs [29].

The fishing down the food web process predicts continuous declines in catches of high trophic level and a shift to small, low-trophic level species [27]. This ubiquitous phenomenon has been identified in several ecosystems, both at local and global scales [30, 39, 67, 95, 96, see more case studies in www.fishingdown.org]. Here, we demonstrate the scenario of overexploitation of high trophic species caught in Arraial do Cabo [54], combining alternative data sources from fishers' LEK and landing data (from 1992 to 2008), supported by other local studies [54]. We observed a declining trend in the mean trophic level and landings of species with TL > 4 and TL ≥ 3.5. When considering the MTL of all species landed, there is no clear pattern in trends, probably due to the decrease in landings of species of low trophic level, such as sardines, and the significant declines in landings of large predators. The same case has been identified in studies in upwelling systems in the Mediterranean Sea [97, 98]. Studies point out that in some cases fishing down may not be identified [39], due to several factors that are responsible for masking the prevalence of this process [39, 99], including taxonomic over-aggregation, the body size of species and geographical expansion of fisheries [33]. Thus, studies suggest the use of MTL as a complementary tool for the assessment of the effects of fishing on ecosystems [39, 100].

Fishers LEK revealed that, in the past, large catches of tunas, sharks and groupers were common in Arraial do Cabo (Fig 6). Shark species such as the spinner shark (*Carcharhinus brevipinna*) were caught in large quantities (Fig 6B), as well as the currently threatened Brazilian guitarfish (*Pseudobatos horkelii*) and the sandbar shark (*Carcharhinus plumbeus*) [54]. Despite never being a major fishing target, these shark species are still caught in the region, but in smaller quantities. The current fishing scenario revealed in the present study is quite different from that reported by the old fishers. Despite its low trophic level, the large body size of parrotfishes such as the endemic greenbeak parrotfish (*Scarus trispinosus*) has made this species a major target of spearfishing, featuring a case of overexploitation and possible local extinction at different parts of the Brazilian coast [54, 101–103]. In Arraial do Cabo, a past study had identified reductions in the abundance and body size of *S. trispinosus* individuals [54]. In addition, spearfishers commonly catch other large parrotfishes as those of the *Sparisoma* genus, indicating shifts in the composition of target species [54].

As we expected, increase in fishing effort was described by fishers as the main cause of overexploitation. While local reports have revealed a declining trend in the number of boats (S2a Fig in S1 Data, S3 Table in S1 Data). Due to the overexploited state of fish stocks, fishers have to spend more time fishing to obtain the same yields as in the past. Another cause of overexploitation reported by fishers was purse seine fisheries. This fishing gear has a high catch potential, and it is less selective when compared to other artisanal fishing gear (e.g. line and hook). For example, in 2019, purse seine catches represented 59% (231 kg) of the total landed catch in the region (S2 Fig in S1 Data). In addition, purse seine effort in fishing days corresponded to only 6% of the total fishing days (S2 Fig in S1 Data). Therefore, despite changes in fishing effort, the scenario of species overexploitation is worrisome and can be even greater than reported in our study.

The few existent management initiatives in the region, which rule over spawning periods of few species, have been ineffective to change the decline trends of fishing resources in Arraial do Cabo. Classical management strategies as minimum landing size of capture, closure periods, fishing quotas with quantities determined after stock assessments should be implemented

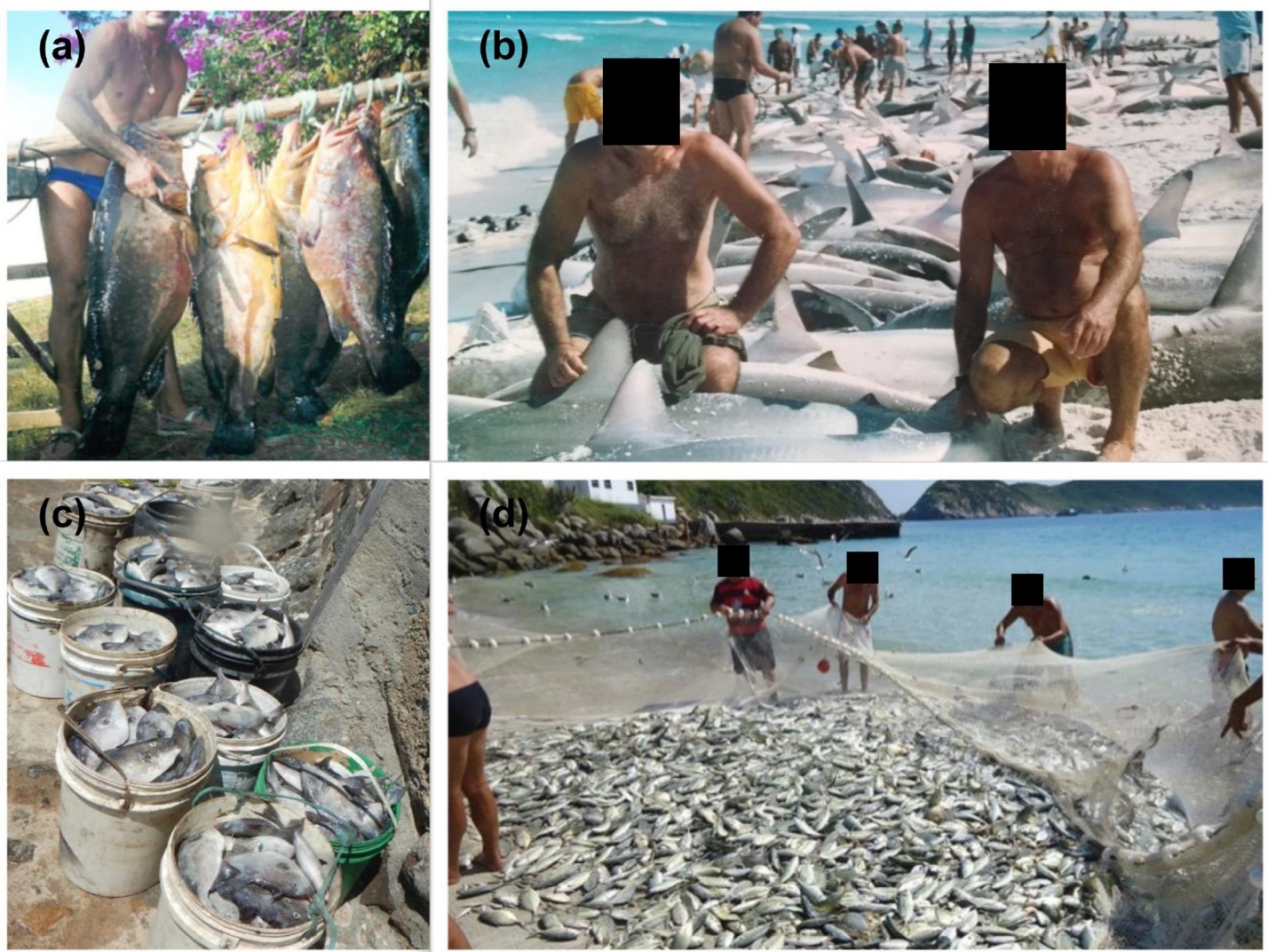

**Fig 6.** Photographs showing past and present catches of species mentioned by fishers as overexploited and new target species in Arraial do Cabo: (a) *Epinephelus marginatus* in 1982; (b) *Carcharhinus brevipinna* caught in Praia Grande in 2005; (c) *Balistes capriscus* and (d) beach seine fishing at Praia Grande recorded in 2020.

based on national conservation status of species. We believe that such strategy may be viable through an adaptive comanagement framework involving stakeholders in all steps of the process. Management may be real-time informed by the small-scale fisheries monitoring that occur in that marine extractive reserve. Threatened species as *E. marginatus*, *Hyporthodus niveatus*, *Lopholatilus villarii*, and *Sparisoma axillare*, which were mentioned by fishers as overexploited and new target species (Tables 1 and 2), should not be targeted until stocks are recovered. As an alternative to minimize illegal fishing off the season and the incidental catch of juveniles, resource users should be oriented to target healthy fish stocks or to search for alternative income sources such as community-based tourism initiatives. A previous study has compared the effects of partially protected areas vs. unprotected areas in the abundance and body size of reef fish, including a site in Arraial do Cabo [104]. Groupers had higher density and relative abundance in a partially protected area, indicating that even small no-take areas can help recover fish populations [104]. The case of Arraial do Cabo is

not different from many reef sites around the world, where the fishing effort has surpassed the recovery potential of natural resources [105]. The good news is that even at relatively small areas groupers can increase in density and size if fishing ceases [104]. Additional ongoing acoustic telemetry study in the region indicate relatively small home ranges (0.05 km$^2$) for two vulnerable parrotfishes (*Sparisoma frondosum* and *S. axillare*) [82], which have implications for establishing the minimum size of no-takes areas (authors, per. com.). Small no-take areas appear as a critical tool to preserve local resources, but more importantly, to show local users their efficiency in rebuilding stocks. Additionally, local fishers can be empowered to explore future established no-take areas through tourism initiatives, having exclusivity over local tourism agencies.

We also highlight the need for greater inspection by the authorities to banning illegal fishing by industrial vessels such as trawl and gillnet fisheries that occur even within the MPA. Encourage dialogue between fishers, especially older and younger generations, so that the elders transmit their knowledge on the past of local fisheries, helping young generations to understand the abundance trends of fish resources. The knowledge of how marine resources were more abundant in the past is linked with higher willingness to collaborate with management and conservation initiatives. This may also influence the overcome of the shifting baseline syndrome among fishers.

Here we demonstrated that experienced fishers better recognize changes in fisheries and in marine ecosystems when compared to less experienced ones. In addition, we identified changes in composition and abundance of pelagic and demersal, reef-associated fish species caught in a secular fishing village using fishers' local ecological knowledge and fisheries landing data. Despite the short time series, fishing monitoring data reveals declining trends in stocks of mesopredators species overtime. The combination of multiple sources of information can help prevent biases as underestimated assessments of the impact of fishing [106]. By incorporating past data, it is possible to identify changes in catches over time, as well as to adjust conservation goals and strategies for marine ecosystems.

## Supporting information

**S1 Data.**
(DOCX)

## Acknowledgments

We thank fishers for their collaboration and shared knowledge. ICMBio has ever been supportiveness for our projects.

## Author Contributions

**Conceptualization:** Carine O. Fogliarini, Vinicius J. Giglio, Mariana G. Bender.

**Formal analysis:** Carine O. Fogliarini.

**Methodology:** Carine O. Fogliarini, Jéssica Bornholdt.

**Writing – original draft:** Carine O. Fogliarini, Carlos E. L. Ferreira, Moysés. C. Barbosa, Vinicius J. Giglio, Mariana G. Bender.

**Writing – review & editing:** Mariana G. Bender.

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
