## [Decision Letter · Decision Letter 0]

19 Nov 2020

PONE-D-20-30952

Fishing down the food web inferred from local ecological knowledge and landing data in the Southeastern Brazilian coast

PLOS ONE

Dear Dr. Oliveira,

Thank you for submitting your manuscript to PLOS ONE. After careful consideration, we feel that it has merit but does not fully meet PLOS ONE’s publication criteria as it currently stands. Therefore, we invite you to submit a revised version of the manuscript that addresses the points raised during the review process.

We look forward to receiving your revised manuscript.

Kind regards,

Hudson Tercio Pinheiro

Academic Editor

PLOS ONE

Journal Requirements:

2.) We note that you have indicated that data from this study are available upon request. PLOS only allows data to be available upon request if there are legal or ethical restrictions on sharing data publicly. For more information on unacceptable data access restrictions, please see http://journals.plos.org/plosone/s/data-availability#loc-unacceptable-data-access-restrictions.

3.) Please amend either the title on the online submission form (via Edit Submission) or the title in the manuscript so that they are identical.

4.) We note that Figure 1 in your submission contain map images which may be copyrighted. All PLOS content is published under the Creative Commons Attribution License (CC BY 4.0), which means that the manuscript, images, and Supporting Information files will be freely available online, and any third party is permitted to access, download, copy, distribute, and use these materials in any way, even commercially, with proper attribution. For these reasons, we cannot publish previously copyrighted maps or satellite images created using proprietary data, such as Google software (Google Maps, Street View, and Earth). For more information, see our copyright guidelines: http://journals.plos.org/plosone/s/licenses-and-copyright.

1.    You may seek permission from the original copyright holder of Figure(s) [#] to publish the content specifically under the CC BY 4.0 license. 

5.) We note that Figure 6 includes images of participants in the study. 

Additional Editor Comments (if provided):

Dear Dr Oliveira,

I received the comments of three reviewers and they recognized the merits of the manuscript. but also recommended major revisions. Please take a look on their comments and suggestions, and describe how you have addressed them in a response letter and a marked version of the manuscript.

I look forward to hearing from you,

Sincerely,

Hudson Pinheiro

Reviewers' comments:

Reviewer's Responses to Questions

**Comments to the Author**

1. Is the manuscript technically sound, and do the data support the conclusions?

Reviewer #1: Yes

Reviewer #2: Partly

Reviewer #3: Yes

2. Has the statistical analysis been performed appropriately and rigorously? 

Reviewer #1: Yes

Reviewer #2: Yes

Reviewer #3: Yes

3. Have the authors made all data underlying the findings in their manuscript fully available?

Reviewer #1: Yes

Reviewer #2: Yes

Reviewer #3: Yes

4. Is the manuscript presented in an intelligible fashion and written in standard English?

Reviewer #1: Yes

Reviewer #2: Yes

Reviewer #3: Yes

5. Review Comments to the Author

Reviewer #1: The manuscript presents a careful analysis and data, although several of them do not present relations as convincing as suggested in the text. However, the manuscript presents so much results and approaches without any novelty for an international audience. It is shown by data from formal science and traditional knowledge that fisheries initially reach the highest trophic levels, that perceptions change over time, that fishing targets are being changed. This is the description of a historical process of any artisanal fisheries in the world. It is not clear what are the innovative aspects in the article that would justify its publication. Considering these aspects, I cannot recommend the publication of the article in a journal such as PLOS ONE

Reviewer #2: Review of de Olivera et al. Fishing down the food web inferred from local ecological knowledge and landing data in the Southeastern Brazilian coast

De Olivera et al used traditional ecological knowledge from fishers in a coastal community of subtropical Brazil (Arraial do Cabo) and historical landing statistics to identify species overexploitation patterns, changes in target fishery resources and trophic levels. Using information from 155 interviews made to fishers and the landing statistics from that area during 1992 and 2008, the authors tried to identify which species according to the perception of fishers were overexploited and which species were new components of the fishery. Additionally, the authors calculated the mean trophic level of the annual landings in that area to verify if the fishing down the food web hypothesis would hold true for that area.

There is a vast amount of literature that has combined TEK with conventional fisheries information to complement our knowledge of a specific fishery. The present study is one of those studies and is a welcome contribution to this relatively new field. I am more critical about how the authors have tried to show their results especially with respect to the fishing down the food web hypothesis and in a lesser degree to the shifting baseline paradigm.

My impression from the data shown is that there is no reason to believe that there is a fishing down the food web pattern in the study system, at least with the information (MTL) that the authors are analyzing. Despite there are a lot of studies that have pointed to a shifting baseline situation in fisheries, were younger fishers won’t recognize easily the degraded state of fishery resources, the present study in my opinion does not show that clear pattern (by inspecting Fig 2a,c). This is totally fine as a result and there is no reason to try to adjust the argument to the prevailing paradigm.

In my opinion, the most interesting finding of the paper is the change in target species pointed out by fishers. This is something that could be highlighted much more in the manuscript. Discussion of the implications of this with similar results in other parts of the world would make the manuscript much stronger.

Having said this, I welcome this contribution to the field, but I disagree with the main interpretation of the data presented in the manuscript. I therefore recommend the authors to reconsider many of the interpretations of their findings in light of what their actual data show. After these major criticisms are addressed I think that this manuscript is not suitable for publication in a PloS ONE.

See specific comments below.

Abstract

L36. Clarify is you are referring to landings or to catches. These are two different concepts. Look at differences here: http://www.fao.org/3/bt981t/bt981t.pdf

L55. Provide a more specific sentence on the problem of sustainability in fisheries. The recent FAO SOFIA reports clearly define and provide a historical perspective on the percentage of fish stocks that are fished at biologically sustainable levels since the 70s.

L61-62. Not all fisheries are directed toward top predators. This is a misleading statement. Modify sentence or bring clarification to what you actually mean.

L71. The “fishing down the food web“ paradigm has been continuously revised since it was first proposed. I would suggest that the authors consider at least mentioning in the introduction the critics of the paradigm. For this, I would recommend to revise Branch et al. (2010) The trophic fingerprint of marine fisheries. Nature 468, 431–435 and Sethi et al (2010) Global fishery development patterns are driven by profit but not trophic level. PNAS 107, 12163-12167.

L73. Change “trophic” for “food”.

L72-79. See comment from L71. Maybe rephrasing the whole paragraph reflecting that there are many ways how fishing can affect marine ecosystems (i.e. not only by reducing trophic level), would be the way to go.

L93. Do you mean unexplored or unexploited?

L109. Change “have” for “has”

L109-114. A number of recent studies have used LEK to link to trophic models. This application may be worth including here and probably somewhere in the discussion. Just a few examples:

Bevilacqua AHV, Carvalho AR, Angelini R, Christensen V (2016) More than Anecdotes: Fishers’ Ecological Knowledge Can Fill Gaps for Ecosystem Modeling. PLoS ONE 11(5): e0155655. https://doi.org/10.1371/journal.pone.0155655

Sánchez-Jiménez A, Fujitani M, MacMillan D, Schlüter A and Wolff M (2019) Connecting a Trophic Model and Local Ecological Knowledge to Improve Fisheries Management: The Case of Gulf of Nicoya, Costa Rica. Front. Mar. Sci. 6:126. doi: 10.3389/fmars.2019.00126

L120. Can you specify what is the MTL approach? Also, I am not sure if you can link possible changes in landings trophic structure exclusively to fishing impacts. There may be other reasons associated these to changes. Provide some clarification on how are you able to assign the possible changes just to fishing.

Materials and methods

L129-130. It is not the artisanal fisheries the one making trips. It is the fishermen those making the trips. Correct wording there.

L131. Clarify the meaning of the word secular in this context.

L151. Snowball sampling has been suggested to sometime bias results of interviews. See:

Kirchherr J, Charles K (2018) Enhancing the sample diversity of snowball samples: Recommendations from a research projecton anti-dam movements in Southeast Asia.PLoSONE 13(8): e0201710.https://doi.org/10.1371/journal.pone.0201710

Explain how do you overcome these limitations or acknowledge the limitations of your strategy.

L155. I would not call a beginner a fisher that has been fishing 15 years. Maybe change the name of this first category.

L156-157. What is the reasoning to differentiate fishers with 36-40yr of experience from those with >40 yr.? I am not arguing for a different categorization, but the reasoning behind this choice needs to be clear.

L173. Landings and catch are two different terms. Be clear in what data you actually obtained. See comment from above.

L186-188. Revise this sentence. It is not clear what you are trying to say.

L196. Is “citations” the right word here?

L212. The normal term is “count data”

Results

L232-239. By looking at the distribution of your data points in Fig2a and 2c it is difficult to find a good fit for any model there. Can you provide a measure of the goodness of fit of your model? Something like AIC or a pseudo-R? include a table with the results of your GLM in the ms or in the Supplement.

L300-302. Fisheries landings data does not necessarily reflect the stock status. Work of Trevor Branch and co-workers among others has repeatedly shown the weaknesses of using catch data to determine stock status. This needs appropriate acknowledgement in the manuscript.

L323-324. This statement is problematic. To my understanding your interviews did not ask about temporal trends to be able to compare those directly with the landings data that you are presenting. Please clarify how you are making these comparisons.

Discussion

L352-353. By looking at your Fig2a and 2c, I would argue that there is no clear pattern to make such an statement.

L396-398. Again, you Fig 5a, does not fully support your statement.

L402-404. From what I got from your methods, you used the landings data to derive mean trophic level of the landings per year. Less clear to me is how you are incorporating LEK into that analysis. Almost all the most important new fishery species identified by the fishers are high trophic level (>4). And the most important, T. lepturus is 4.4.

L409-416. This sounds as if you would like to prove that fishing down the food webs is occurring in your study site, but your data does not tell that is actually happening. You provide alternative explanations from other areas but don’t provide an explanation or analysis for your case.

L418-419. Fig 6 show very impressive pictures, but your LEK study did not ask questions about tunas or sharks and their historical catches.

L425-430. Why Scarus trispinosus did not appear in your LEK results? Fishers did not recognize this species as overexploited. Also this whole section does not discuss anything related to your study.

L436-440. By looking at your supplement data, the number of fishing boats has been reduced to around half of those reported in 1995. Fishing days in contrast have more than doubled. The average number of fishing days per boat per year in 1995 was only 5 days. In 2019 this value is 25 days per year. The 25 days value is still very low for a year. What are the explanation that you have for this? Are the methodologies to calculate these numbers comparable among reports?

L448. In the manuscript you do not present any information about those management initiatives and how you relate those to your actual work

L450. What is the likelihood that fishing quotas are going to be enforced in such areas? Is this a realistic strategy?

L469-471. This argument is counter-intuitive. Your manuscript is about using LEK to show the exploitation status of fisheries resources. And your recommendation now is to educate those fishers because they do not know about the status of their resources.

useful reference missing in the ms:

Silvano, R.A.M. and Begossi, A., 2010. What can be learned from fishers? An integrated survey of fishers’ local ecological knowledge and bluefish (Pomatomus saltatrix) biology on the Brazilian coast. Hydrobiologia, 637(1), p.3.

Reviewer #3: Fogliarini et al., is an interesting ms dealing with the fishing down the food web process inferred from local ecological knowledge and landing data. The ms is pleasant and the ideas are organized and written with high quality. I have included a series of comments throughout the ms that I would like to see addressed before publication.

One important concern is regarding the different time scale from LEK data and from landing data. Interviews were performed from 2018 to 2019 and fishing land data from 1992 to 2008. there is at least a decade temporal difference on this data. Authors are confident that this do not imply results bias on the present study? I would like to see the development of this controversial time different in the ms discussion.

Results are well-presented and consistent. However, figures are verry low quality and must be redone. See comments on the text. Fig 1 (map) is extremely low quality and unsuitable for publication.

Discussion is organized but lack of details in some important information. Comments are in the attached PDF.

I would be keen to review the ms once again before publication.

6. PLOS authors have the option to publish the peer review history of their article (what does this mean?). If published, this will include your full peer review and any attached files.

Reviewer #1: No

Reviewer #2: No

Reviewer #3: No

---

## [Author Response · Author response to Decision Letter 0]

22 Jan 2021

Response to Reviewers

Reviewer #1: The manuscript presents a careful analysis and data, although several of them do not present relations as convincing as suggested in the text. However, the manuscript presents so much results and approaches without any novelty for an international audience. It is shown by data from formal science and traditional knowledge that fisheries initially reach the highest trophic levels, that perceptions change over time, that fishing targets are being changed. This is the description of a historical process of any artisanal fisheries in the world. It is not clear what are the innovative aspects in the article that would justify its publication. Considering these aspects, I cannot recommend the publication of the article in a journal such as PLOS ONE

R. We respect Reviewer #1 for his opinion and understand that this is a common pattern reported for artisanal fisheries across the world. Yet there are few studies in which Local Ecological Knowledge, from traditional communities, has been contrasted with landing data to reveal matching patterns in local fisheries (Silvano et al. 2009, Ferreira et al. 2014, Herbst & Hanazaki, 2014, Silva et al. 2020). Considering this aspect, at the best of our knowledge, the present study is the first to assess exploitation status of fish species caught by artisanal fishing and changes in target species over time in the southwestern Atlantic. We highlight that the study area is a partially-protected marine area, whose main resource managers are the extractive community. Here, we combine the knowledge of fishers with landings data, to understand the dynamics of fish stocks from the past to the present. Thus, we generate information that can be used to propose conservation and management measures. We point out that there is no information of temporal changes in fishery targets along Brazilian coast. Such information is not only of ecological importance, but also cultural and economic. Also, there are several examples of local scale studies published in PLOS ONE, that are not necessarily of broad interest (ttps://journals.plos.org/plosone/article?id=10.1371/journal.pone.0236146)

(https://journals.plos.org/plosone/article?id=10.1371/journal.pone.0228976) 

Reviewer #2: Review of de Oliveira et al. Fishing down the food web inferred from local ecological knowledge and landing data in the Southeastern Brazilian coast

De Oliveira et al used traditional ecological knowledge from fishers in a coastal community of subtropical Brazil (Arraial do Cabo) and historical landing statistics to identify species overexploitation patterns, changes in target fishery resources and trophic levels. Using information from 155 interviews made to fishers and the landing statistics from that area during 1992 and 2008, the authors tried to identify which species according to the perception of fishers were overexploited and which species were new components of the fishery. Additionally, the authors calculated the mean trophic level of the annual landings in that area to verify if the fishing down the food web hypothesis would hold true for that area.

There is a vast amount of literature that has combined TEK with conventional fisheries information to complement our knowledge of a specific fishery. The present study is one of those studies and is a welcome contribution to this relatively new field. I am more critical about how the authors have tried to show their results especially with respect to the fishing down the food web hypothesis and in a lesser degree to the shifting baseline paradigm.

My impression from the data shown is that there is no reason to believe that there is a fishing down the food web pattern in the study system, at least with the information (MTL) that the authors are analyzing. Despite there are a lot of studies that have pointed to a shifting baseline situation in fisheries, where younger fishers won’t recognize easily the degraded state of fishery resources, the present study in my opinion does not show that clear pattern (by inspecting Fig 2a,c). This is totally fine as a result and there is no reason to try to adjust the argument to the prevailing paradigm.In my opinion, the most interesting finding of the paper is the change in target species pointed out by fishers. This is something that could be highlighted much more in the manuscript. Discussion of the implications of this with similar results in other parts of the world would make the manuscript much stronger.

Having said this, I welcome this contribution to the field, but I disagree with the main interpretation of the data presented in the manuscript. I therefore recommend the authors to reconsider many of the interpretations of their findings in light of what their actual data show. After these major criticisms are addressed I think that this manuscript is not suitable for publication in a PloS ONE.

See specific comments below.

R: We agree that the fishing down the food web process was not detected when we analyzed the MTL of all landed species. However, this process may not be easily detected due to the influence of factors that may mask its occurrence. Some factors that may mask fishing down are taxonomic over aggregation of catch statistics and the spatial expansion of fishing, but that can be clarified by local experts. Also, the masking effect can occur when both the proportion of high TL species and pelagic species of low TL are declining. In our case, we identified declines in landings of the two main landed species, largehead hairtail (Trichiurus lepturus) and the sardine (Sardinella brasiliensis) (see figures below). Landings of T. lepturus, a high TL species, had an increase from the 1990s to 2000s, followed by a decrease in landings after the 2000s. The same decline trend was identified for the sardine, the main species landed in the region. Declining trends in landings of these species are possibly causing an increase in the MTL of all species, as demonstrated in other studies (Blanchard et al. 2010, Coll et al. 2010). In this case, the masking effect is named as “false fishing up” and we discussed the possible occurrence of this process in our study (lines 440 - 447). Another important result that suggests the overexploitation scenario was showed through landings of high TL species (TL> 4) that decreased significantly over time (Fig. 5c, d). For some stocks, the reduction exceeded 80% (Fig. 4a, c, e, g). In addition, we asked fishers about the status of fishing stocks, and the answers to this question were not added to our manuscript, but it is important to highlight that 92% of fishers reported a decline in fishing stocks, and most of them reported that fish stocks decrease from 50 to 80% (see figure below). Finally, the overexploitation scenario of marine resources has been supported by another local study (Bender et al. 2014) based on fisher’s knowledge, underwater visual census and scientific data. The authors also identified declines in high and low TL species caught by artisanal fisheries.

With respect to the shifting baseline results, we understand the reason for his argument. The distribution of the number of citations does not vary so much between years of fishing practice (fig. 2 a, c). However, we showed that there were different percent increase in citations as fishers’ experience increase (Fig. 2 b, d). Another evidence of the shifting baseline occurs when there are significant differences in the proportion of citations between experience categories (Fig. 3). Very experienced fishers identified Epinephelus marginatus, Caranx hippos, and Balistes capriscus as overexploited and new target species more frequently than the less experienced ones (Fig. 3).

Abstract

L36. Clarify is you are referring to landings or to catches. These are two different concepts. Look at differences here: http://www.fao.org/3/bt981t/bt981t.pdf

R: Thank you for your comment. We changed this expression along to manuscript. In our case, the correct term is "landings" because it represents the quantities landed as recorded at the time of landing as whole or eviscerated fish, fillets, livers, roes, etc.

L55. Provide a more specific sentence on the problem of sustainability in fisheries. The recent FAO SOFIA reports clearly define and provide a historical perspective on the percentage of fish stocks that are fished at biologically sustainable levels since the 70s.

R: We added the following sentence: “The status assessment of fishery resources revealed that the fraction of fish stocks that are within biologically sustainable levels decreased from 90% in 1974 to 65.8% in 2017 (FAO, 2020).”

L61-62. Not all fisheries are directed toward top predators. This is a misleading statement. Modify sentence or bring clarification to what you actually mean.

R: We agree with your statement and modified the sentence (Lines 62 - 65).

L71. The “fishing down the food web“ paradigm has been continuously revised since it was first proposed. I would suggest that the authors consider at least mentioning in the introduction the critics of the paradigm. For this, I would recommend to revise Branch et al. (2010) The trophic fingerprint of marine fisheries. Nature 468, 431–435 and Sethi et al (2010) Global fishery development patterns are driven by profit but not trophic level. PNAS 107, 12163-12167.

R: We have included the following explanation: “In addition, studies criticize the exclusive use of MTL as an indicator to track changes in the ecosystem (Branch et al. 2010). As well as there may be multiple hypotheses (Branch et al. 2015), which explain how fishing affects marine food webs, such as profit-driven fishing (Sethi et al. 2010).”

L73. Change “trophic” for “food”.

R: The term was replaced. (Line 74).

L72-79. See comment from L71. Maybe rephrasing the whole paragraph reflecting that there are many ways how fishing can affect marine ecosystems (i.e. not only by reducing trophic level), would be the way to go.

R: We agreed and added an explanation in lines 73 - 84.

L93. Do you mean unexplored or unexploited?

R: We mean "unexplored state". We clarified it in the text.

L109. Change “have” for “has”

R: Replaced accordingly. 

L109-114. A number of recent studies have used LEK to link to trophic models. This application may be worth including here and probably somewhere in the discussion. Just a few examples:

Bevilacqua AHV, Carvalho AR, Angelini R, Christensen V (2016) More than Anecdotes: Fishers’ Ecological Knowledge Can Fill Gaps for Ecosystem Modeling. PLoS ONE 11(5): e0155655. https://doi.org/10.1371/journal.pone.0155655

Sánchez-Jiménez A, Fujitani M, MacMillan D, Schlüter A and Wolff M (2019) Connecting a Trophic Model and Local Ecological Knowledge to Improve Fisheries Management: The Case of Gulf of Nicoya, Costa Rica. Front. Mar. Sci. 6:126. doi: 10.3389/fmars.2019.00126

R: We added these studies in lines 112 - 116.

L120. Can you specify what is the MTL approach? 

R. Done. Lines 123 - 124.

Also, I am not sure if you can link possible changes in landings trophic structure exclusively to fishing impacts. There may be other reasons associated these to changes. Provide some clarification on how are you able to assign the possible changes just to fishing.

R: First, we observed through information from fishers that a large part of the resources identified as overexploited were high trophic level species. This finding motivated us to try to understand how fishing is affecting the local ecosystem. Second, the temporal replacement of target species caught our attention –as the case of largehead hairtail (T. lepturus)– which became a new resource, and posteriorly, revealed declining trends. Finally, to track changes in the trophic structure, we used landing data to calculate the mean trophic level index of landed species. We verified MTL trends in four groups: all landed species, species with TL > 4, TL ≥ 3.5 and TL < 3.5. We identified significant decline trends in MTL of species with TL > 4 and TL ≥ 3.5, while the MTL of all landed species and TL < 3.5 showed non-significant increase trends. Although the MTL of all species had no decline, we noticed that when the sardine and largehead hairtail are removed, there is a decline over time. The sardine is the main species landed in the period, followed by P. saltatrix and T. lepturus. The approach of excluding the sardine (low TL) and T. lepturus (high TL) from the is based on previous studies (Alleway et al. 2014). Studies have shown that fishing down can be masked due to declines in small pelagic species with low TL and marked by declines of large predators in the past (Blanchard et al. 2010, Coll et al. 2010). We suggest that this masking effect is a result from the overexploitation of sardine and T. lepturus.

Also, we checked the landing trends for species reported by fishers as overexploited and new targets. This approach helped us to identify significant declines in landings of species with high trophic levels (TL = 4 and TL ≥ 3.5) (figure 5a, b). In addition, the declining trends for high trophic level species were also observed through landings data (Kg) (figure 5c), and separately for the main species landed (figure 5d). 

Materials and methods

L129-130. It is not the artisanal fisheries the one making trips. It is the fishermen those making the trips. Correct wording there.

R: Corrected accordingly (Lines 133 - 135).

L131. Clarify the meaning of the word secular in this context.

R: Secular means that this fishing activity has been developed for over a century in the region. Studies show that beach seine fishing is practiced with old fishing canoes, which were built by the first generations of fishers in Arraial do Cabo, since 1880. This information is a book entitled “Arraial 500” and was clarified in the text (Line 135).

L151. Snowball sampling has been suggested to sometime bias results of interviews. See:

Kirchherr J, Charles K (2018) Enhancing the sample diversity of snowball samples: Recommendations from a research projecton anti-dam movement in Southeast Asia. Plos ONE 13(8): e0201710.https://doi.org/10.1371/journal.pone.0201710

Explain how do you overcome these limitations or acknowledge the limitations of your strategy.

R: In fact, we used snowball sampling to complete our sample number. At the fishing pier, we approached fishers at random, and they were invited to participate in our research. In addition, we had the help of an experienced fisher known to the community. With this contact, it was possible to access a large number of interviewees and it mainly facilitated the process of interviews. Kirchherr & Charles (2018) recommend strategies to overcome snowball sampling limitations, as having previous personal contacts before starting snowball sampling. We therefore did not rely exclusively on snowball sampling.

L155. I would not call a beginner a fisher that has been fishing 15 years. Maybe change the name of this first category.

R: Thank you for the suggestion. We modified that term into “less experienced”.

L156-157. What is the reasoning to differentiate fishers with 36-40yr of experience from those with >40 yr.? I am not arguing for a different categorization, but the reasoning behind this choice needs to be clear.

R: We chose these experience categories to accommodate similar sample sizes. Thus, there is a better distribution of the number of interviewees in each experience category. In addition, experience categories were based on previous studies (e.g. Maia et al. 2018). We have included an explanation in lines 164 – 173. 

L173. Landings and catch are two different terms. Be clear in what data you actually obtained. See comment from above.

R: We have replaced “catches” by “landings”. 

L186-188. Revise this sentence. It is not clear what you are trying to say.

R: We have rephrased this fragment, see lines 196 – 198.

L196. Is “citations” the right word here?

R: Yes, we consider that the correct form is "citations" because it is the frequency at which a species was mentioned by the fishers as overexploited and new target species.

L212. The normal term is “count data”

R: Done. Line 224.

Results

L232-239. By looking at the distribution of your data points in Fig2a and 2c it is difficult to find a good fit for any model there. Can you provide a measure of the goodness of fit of your model? Something like AIC or a pseudo-R? include a table with the results of your GLM in the ms or in the Supplement.

R: As suggested by the reviewer, we added a table with the results of GLM (Please, see table 1). 

L300-302. Fisheries landings data does not necessarily reflect the stock status. Work of Trevor Branch and co-workers among others has repeatedly shown the weaknesses of using catch data to determine stock status. This needs appropriate acknowledgement in the manuscript.

R: We agree with your comment, the study by Branch and collaborators was included in lines 80 - 83, this point was also argued in lines 439 – 448, in which we discussed that MTL index may not always reveal the true status of stocks. However, our work uses as a prior knowledge the information from local fishers, who reported that overexploited species such as Pomatomus saltatrix and Trichiurus lepturus, among many others, species whose declines have also been evidenced through landing data. In general, when fishers were asked about the status of fishing stocks, 92% of fishers reported a decline in stocks of target species. In addition, fishers were asked about the percentage of declines, and the highest percentages are between 50% and 80% (see figure in the answer above). This finding, added to the declining trends in landings of high trophic level species suggest the overexploitation scenario of fisheries in Arraial do Cabo. Old photographs help fishers’ remember the past composition and abundance of landed species, and visualize change in fisheries (Fig. 6). These photographs reveal catches with large predators such as groupers and sharks, which have become rare events in the region. Photographs can be valuable historical information sources. In previous studies, old photographs were used to estimate decreases in the maximum individual fish size and the proportion of large groupers caught in South Florida, USA (McClenachan, 2009). 

L323-324. This statement is problematic. To my understanding your interviews did not ask about temporal trends to be able to compare those directly with the landings data that you are presenting. Please clarify how you are making these comparisons.

R: Thank you for your comment. Indeed our interviews did not ask fishers on the past vs. present day catches of each species, but solely on which fish species they considered overexploited, and which were new target species, based on their accumulated experiences. Our questionnaire was organized to detect overexploited and new target species, which were then compared to temporal trends in local landings. We understand that landing trends are similar to the species identified as overexploited by fishers, where we can see declines in landings, and also for new target species, for which landings increase (as they become new targets) and are followed by a decrease. 

Discussion

L352-353. By looking at your Fig2a and 2c, I would argue that there is no clear pattern to make such a statement.

R: This sentence refers to the increase in the number of species reported by fishers as years of practice increase. We identified this pattern in figure 2a, c and through the GLM results, which show significant increases in the number of species cited as overexploited (P = 0.003) as well as for the number of species identified as new target species (P = 0.014) across years of practice. We understand that the variance around the number of species mentioned by fishers across experience categories gives a sense of no clear pattern, yet our model has revealed there are slight increases across each category.

L396-398. Again, your Fig 5a, does not fully support your statement.

R: We agree that figure 5a does not fully support our argument for the occurrence of this phenomenon in Arraial do Cabo. For clarification, we have corrected this statement in the fragment (lines 429 - 432) and throughout the manuscript. We also included a case study that showed the change in target species from demersal finfish to shellfish, which identified the fishing down process through a long series of landing data (1920 - 2010) and MTL. In our study, we used a short series of landing data (1992 - 2008) in combination with fisher’s knowledge. Despite the short series of landing data, we interviewed 32 very experienced fishers (with practice time between 40 - 60 years) who also reported declines in fish stocks. In addition, landing data show significant declines in species with TL > 4 (Fig. 5c, d). Finally, we are cautious in affirming the occurrence of fishing down in Arraial do Cabo, although there is the possibility that it may be masked by other issues.

L402-404. From what I got from your methods, you used the landings data to derive mean trophic level of the landings per year. Less clear to me is how you are incorporating LEK into that analysis. Almost all the most important new fishery species identified by the fishers are high trophic level (>4). And the most important, T. lepturus is 4.4.

R: We agree that most of the new target species reported by fishers are of a high trophic level, such as T. lepturus (TL = 4.4) and B. capriscus (TL = 4.1). The fisher’s knowledge helped us to identify the declines in top predators such as Pomatomus saltatrix (TL = 4.5), Trichiurus lepturus (TL = 4.4), Epinephelus marginatus (TL = 4.4), Seriola fasciata (TL = 4.5), and Caranx hippos (TL = 3.6). Yet, fishers reported a scenario of overfishing in the region, which 92% of them noticed alarming declines in stocks of the main target species of fishing. Another way to incorporate the fisher’s knowledge in our analysis was comparing the MTL of overexploited with MTL of new target species mentioned by fishers (see Supporting information, Fig. S1). We observed that the citations of overexploited species are more frequently within higher TL species than new target species (see Supporting information, Fig. S1). In addition, the MTL observed for overexploited species was higher than expected at random. 

L409-416. This sounds as if you would like to prove that fishing down the food webs is occurring in your study site, but your data does not tell that is actually happening. You provide alternative explanations from other areas but don’t provide an explanation or analysis for your case. 

R: Thank you for your comment. We have rewritten this paragraph (Lines 439 - 448).

L418-419. Fig 6 show very impressive pictures, but your LEK study did not ask questions about tunas or sharks and their historical catches.

R: The shark species of the Carcharhinus genus have been cited by fishers as overexploited and new target species (see tables 2 and 3), as it is the case for C. brevipinna (Figure 6). Our mention of shark and tuna catches merely intended to offer the reader an insight into catches' composition at different time periods, as previous studies have identified declines in large predators in the region (Bender et al. 2014). 

L425-430. Why Scarus trispinosus did not appear in your LEK results? Fishers did not recognize this species as overexploited. Also this whole section does not discuss anything related to your study.

R: Thank you for your comment. S. trispinosus did not appear in our results because this species is mainly a target of spearfishers. For this fishing gear, we interviewed two local fishers, and one of them mentioned the species Sparisoma axillare as a new target species. Another point that explains why this species was not mentioned by fishers is that, unlike in our study, Bender et al. (2014) asked questions directed towards parrotfish and grouper species, using photographs. Indeed, the collapse of parrotfishes in the same region is pretty well covered in Bender et al. (2014).

L436-440. By looking at your supplement data, the number of fishing boats has been reduced to around half of those reported in 1995. Fishing days in contrast have more than doubled. The average number of fishing days per boat per year in 1995 was only 5 days. In 2019 this value is 25 days per year. The 25 days value is still very low for a year. What are the explanation that you have for this? Are the methodologies to calculate these numbers comparable among reports?

R: We agree with your comment and removed that sentence (lines 468 - 471). In figure and table S2, the increase in fishing days corresponds to the most recent reports from PMAP-BS, both accounting for the fishing days of all fishing gear operating in Arraial do Cabo. These recent reports also consider gear that does not use boats, e.g. hand line, manual collection and some types of fishing nets, as beach seine. Although these reports used different methodologies to estimate the number of fishing days, we can still consider the number of boats as the fishing effort to compare the causes reported by fishers.

L448. In the manuscript you do not present any information about those management initiatives and how you relate those to your actual work

R: The only management initiatives existent in the local marine extractive reserve is regarding the spawning period for some species and few species prohibition. Indeed, there are no initiatives for stock recovering based on data produced locally. Additionally, local managers are poorly skilled to absorb state and national management rules to be locally applied and enforced. 

L450. What is the likelihood that fishing quotas are going to be enforced in such areas? Is this a realistic strategy?

R: Yes, it is a realistic strategy since fishing is practiced in a relatively small partially protected MPA. Also, this MPA has financial and logistic resources to implement a collaborative co-management framework involving stakeholders and being informed by the small-scale fishing monitoring conducted by the state fishery agency. We clarified in the text (lines 480 - 485).

L469-471. This argument is counter-intuitive. Your manuscript is about using LEK to show the exploitation status of fisheries resources. And your recommendation now is to educate those fishers because they do not know about the status of their resources.

R: We agree with your suggestion and have rewritten this sentence. (Lines 507 - 512).

Useful reference missing in the ms:

Silvano, R.A.M. and Begossi, A., 2010. What can be learned from fishers? An integrated survey of fishers’ local ecological knowledge and bluefish (Pomatomus saltatrix) biology on the Brazilian coast. Hydrobiologia, 637(1), p.3.

R: We added this reference in the introduction section. 

Reviewer #3: Fogliarini et al., is an interesting ms dealing with the fishing down the food web process inferred from local ecological knowledge and landing data. The ms is pleasant and the ideas are organized and written with high quality. I have included a series of comments throughout the ms that I would like to see addressed before publication.

One important concern is regarding the different time scale from LEK data and from landing data. Interviews were performed from 2018 to 2019 and fishing land data from 1992 to 2008. there is at least a decade temporal difference on this data. Authors are confident that this do not imply results bias on the present study? I would like to see the development of this controversial time different in the ms discussion.

Results are well-presented and consistent. However, figures are very low quality and must be redone. See comments on the text. Fig 1 (map) is extremely low quality and unsuitable for publication.

Discussion is organized but lack of details in some important information. Comments are in the attached PDF.

I would be keen to review the ms once again before publication.

R. We thank Reviewer #3 for his/her comments and suggestions. We have tried to include all of the proposed alterations in this new MS version. We understand there is a time lag from landing data and the time when interviews were conducted but we are also certain that this does not affect our MS main conclusions. If the current time difference already reveals concordant patterns between landings trends and species mentioned as overexploited (declining trends) and new target species (increasing), we believe that if a longer landings data-series was available, that would make the patterns even clearer. If fishers were interviewed back in 2008, they would have possibly given the same answers on the current new target species and overexploited ones (see Bender et al. 2014 for overexploited taxa). We tried to have access to a longer landing data-series but did not receive any answers from local authorities that are responsible for fisheries monitoring. Data exists but won't be made available. We must also point out that targeted species have remained the same for the last 20 years (CEL Ferreira). 

Major comments reviewer #3:

L46. ecological conclusion?? It is not a methodological paper...

The study adds extremely relevant information on the overexploitation status of many different species…etc.

R: We agree and modified this sentence into: "Our work reveals a depletion scenario of fishing stocks and changes in composition of fish species caught in the Arraial do Cabo.”

L55. Fishing is not only an impact. Several fishing communities live from those resources and cope with the human X ocean relationship. Important to point out that unregulated and predatory fishing is the problem addressed herein.

R: Thank you for your comment. We agreed and modified this sentence. See line 54.

L62. Briefly mention the consequences please.

R: Among the consequences of unsustainable removal of top predators are the release of prey. In other words, the loss of top predators can cause numerical increases in meso-consumers and alter community structure. We mention other consequences in the following paragraph. Please see line 65.

L79.“....have been proposed so far and our study aims to add information on...”

Complete your sentence as suggested so it would be more useful.

R: This sentence was rewritten (Lines 79 - 84).

L126. Poor figure quality. Difficult to identify where is study area in Brazilian coast. 

I would like to suggest a new map with better quality.

R: We adjust the quality of figures.

L132. Not mentioned in the Fig where is the MPA.

R: The figure 1 was substitute by an adapted map due copyright for publication, in this new version there is no mention of the MPA.

L165. This seems like a discussion. Any reference for this idea?

R: We added the reference Mendonça et al. (2013).

L175. Interviews were acquired from 2018 to 2019 and fishing land data from 1992 to 2008. there is at least a decade temporal difference on this data. Authors are confident that this do not imply results bias on the present study? I would like to see the development of this controversial time different in the ms discussion.

R: Please see answer above (L180). We have also included that in the discussion. 

L176. In the reference list? How this data was acquired?

R: It is not in the list of references because these data were acquired directly from the Fundação de Pesca de Arraial do Cabo - FIPAC. These is the first fisheries monitoring reports in Arraial do Cabo, and are available on the FIPAC website (see https://www.fipac.rj.gov.br).

L180. Again. As previously mentioned, this is a quite old data. Fishing data from 1992 (almost 30 years old) represents the MTL current scenario? Probably trophic levels are even worst nowadays. This must be discussed.

R: We agree with reviewer’s comment that there are temporal differences in the data sources. Fisheries statistics were obtained through Fundação Instituto da Pesca de Arraial do Cabo (FIPAC) which shared only a short monitoring time series (1992 – 2008). We agree that the current scenario for trophic levels can be even worse, as our results showed declines in high-trophic species. 

L346. This result came out of nowhere. Please, describe in the aims, methods etc.

R: We describe this result in all sections of our manuscript. Please see how we described in each section:

Introduction (Objectives): “We also discuss the main causes of fish overexploitation and management strategies in a subtropical fisheries hotspot.”

Materials and methods (Data collection): “(iii) what are the main causes

of species’ overexploitation”.

Results: “Fishers mentioned 35 different possible causes of fisheries overexploitation in Arraial do Cabo (1.58 ± 1.08, mean ± standard deviation). The five most mentioned causes were the increased fishing effort (22%), which is recognized by fishers as an increase in the number of fishers and boats; presence of industrial fishing vessels (21%), presence of purse seine fisheries (19%), gillnet fishing (16%), and trawling (12%).”

In the following paragraph of the discussion: “The cause more mentioned by fishers as responsible for fisheries overexploitation was increase in the fishing effort…”

L366. where are the ecological implications? this paragraph needs to end up with a broad view of your main results.

R: We agree with your suggestion and incorporated more explanations in this paragraph. Please see lines 396-400.

L465. how small? Is a no-take zone implementation process been currently discussed with local community?

R: We correct the term “small MPAs” to “small no-take areas”. The implementation process had been discussed with the local community recently. In 2021 a management plan with the contribution of researchers, community members, and fishers, will suggest the implementation of 3 to 4 small no-take areas within the extractive marine reserve. It is a participatory process, which will establish specific rules for the use of each of these areas.

L465. how short? More info is necessary.

R: We add more explanation in lines 497 - 500.

L480. I would like to see more discussion on the conservation strategies along the ms discussion.

R: We added more details on the management strategies, following the reviewers’ suggestion. Now, we describe, minimum landing size of capture, closure periods how fishing quotas can be implemented in a participatory manner the importance of no-take MPAs, a improve of surveillance for banning the illegal fisheries and the importance of young generations acknowledging the past abundance of marine resources to be more willing to accept management and conservation initiatives (lines 478 - 504). 

References 

Alleway HK, Connell SD, Ward TM, & Gillanders BM. (2014). Historical changes in mean trophic level of southern Australian fisheries. Marine and Freshwater Research, 65(10), 884-893.

Bender MG, Machado GR, de Azevedo Silva PJ, Floeter SR, Monteiro-Netto C, Luiz OJ, & Ferreira CE. (2014). Local ecological knowledge and scientific data reveal overexploitation by multigear artisanal fisheries in the Southwestern Atlantic. PLoS One, 9(10), e110332.

Blanchard JL, Coll M, Trenkel VM, Vergnon R, Yemane D, Jouffre D, et al. (2010). Trend analysis of indicators: a comparison of recent changes in the status of marine ecosystems around the world. ICES J Mar Sci. 67 (4): 732-744.

Branch et al. (2010). The trophic fingerprint of marine fisheries. Nature 468, 431–435.

Branch TA. (2015). Fishing Impacts on Food Webs: Multiple Working Hypotheses. 576 Fisheries. 40 (8): 373-375.

Coll M, Shannon LJ, Yemane D, Link JS, Ojaveer H, Neira S, et al. (2010). Ranking the ecological relative status of exploited marine ecosystems. ICES J Mar Sci. 67 (4): 769-786.

Da Silva, L. D. M. C., Machado, I. C., dos Santos Tutui, S. L., & Tomás, A. R. G. (2020). Local ecological knowledge (LEK) concerning snook fishers on estuarine waters: Insights into scientific knowledge and fisheries management. Ocean & Coastal Management, 186, 105088.

FAO. 2020. The State of World Fisheries and Aquaculture 2020. Sustainability in action. Rome. https://doi.org/10.4060/ca9229en

Ferreira, H. M., Reuss-Strenzel, G. M., Alves, J. A., & Schiavetti, A. (2014). Local ecological knowledge of the artisanal fishers on Epinephelus itajara (Lichtenstein, 1822) (Teleostei: Epinephelidae) on Ilhéus coast–Bahia State, Brazil. Journal of ethnobiology and ethnomedicine, 10(1), 51.

Herbst, D. F., & Hanazaki, N. (2014). Local ecological knowledge of fishers about the life cycle and temporal patterns in the migration of mullet (Mugil liza) in Southern Brazil. Neotropical Ichthyology, 12(4), 879-890.

Kirchherr, J., Charles, K. (2018). Enhancing the sample diversity of snowball samples: Recommendations from a research projecton anti-dam movement in Southeast Asia. Plos ONE 13(8): e02017

McClenachan, L. (2009). Historical declines of goliath grouper populations in South Florida, USA. Endangered Species Research, 7(3), 175-181.

Sethi et al. (2010). Global fishery development patterns are driven by profit but not trophic level. PNAS 107, 12163-12167.

Silvano, R. A. M., & Begossi, A. (2010). What can be learned from fishers? An integrated survey of fishers’ local ecological knowledge and bluefish (Pomatomus saltatrix) biology on the Brazilian coast. Hydrobiologia, 637(1), 3.

---

## [Decision Letter · Decision Letter 1]

7 Apr 2021

PONE-D-20-30952R1

Telling the same story: fishers and landing data reveal changes in fisheries on the Southeastern Brazilian coast

PLOS ONE

Dear Dr. Oliveira,

Thank you for submitting your manuscript to PLOS ONE. After careful consideration, we feel that it has merit but does not fully meet PLOS ONE’s publication criteria as it currently stands. Therefore, we invite you to submit a revised version of the manuscript that addresses the points raised during the review process.

We look forward to receiving your revised manuscript.

Kind regards,

Hudson Tercio Pinheiro

Academic Editor

PLOS ONE

Journal Requirements:

Additional Editor Comments (if provided):

Dear authors,

Thank you for successfully improve the manuscript following the reviewers suggestions. One of the reviewers made few minor comments in the attached PDF, in the highlighted ms section. I will consider accepting the manuscript after these minor revisions.

Sincerely yours,

Hudson Pinheiro

Reviewers' comments:

Reviewer's Responses to Questions

**Comments to the Author**

1. If the authors have adequately addressed your comments raised in a previous round of review and you feel that this manuscript is now acceptable for publication, you may indicate that here to bypass the “Comments to the Author” section, enter your conflict of interest statement in the “Confidential to Editor” section, and submit your "Accept" recommendation.

Reviewer #3: All comments have been addressed

2. Is the manuscript technically sound, and do the data support the conclusions?

Reviewer #3: Yes

3. Has the statistical analysis been performed appropriately and rigorously? 

Reviewer #3: Yes

4. Have the authors made all data underlying the findings in their manuscript fully available?

Reviewer #3: Yes

5. Is the manuscript presented in an intelligible fashion and written in standard English?

Reviewer #3: Yes

6. Review Comments to the Author

Reviewer #3: Authors have dramatically improved the ms and all the suggestions pointed by reviews have been addressed. I commend the authors for their effort.

Just minor comments on the attached PDF.

7. PLOS authors have the option to publish the peer review history of their article (what does this mean?). If published, this will include your full peer review and any attached files.

Reviewer #3: **Yes: **Pedro Henrique Cipresso Pereira

---

## [Author Response · Author response to Decision Letter 1]

9 Apr 2021

In this version of the manuscript, following the recommendation made by editor and reviewers, we altered lines 458 - 459 and included the recent work of Pereira et al. (2021)* in the reference list. We hope that this new version of the manuscript will be in accordance with the recommendations and scope of Plos One.

*Pereira PHC, Ternes MLF, Nunes JAC, Giglio VJ. Overexploitation and behavioral changes of the largest South Atlantic parrotfish (Scarus trispinosus): Evidence from fishers' knowledge. Biol Conserv. 2021; 254, 108940”.

---

## [Editor Report · Decision Letter 2]

17 May 2021

Telling the same story: fishers and landing data reveal changes in fisheries on the Southeastern Brazilian coast

PONE-D-20-30952R2

Dear Dr. Oliveira,

We’re pleased to inform you that your manuscript has been judged scientifically suitable for publication and will be formally accepted for publication once it meets all outstanding technical requirements.

Kind regards,

Hudson Tercio Pinheiro

Academic Editor

PLOS ONE

Additional Editor Comments (optional):

Dear authors,

Thank you for adjusting the manuscript following the reviewers comments. The paper is now accepted for publication,

Sincerely,

Hudson Pinheiro
---

## [Editor Report · Acceptance letter]

21 May 2021

PONE-D-20-30952R2 

Telling the same story: fishers and landing data reveal changes in fisheries on the Southeastern Brazilian Coast 

Dear Dr. Fogliarini:

I'm pleased to inform you that your manuscript has been deemed suitable for publication in PLOS ONE. Congratulations! Your manuscript is now with our production department. 

Kind regards, 

on behalf of

Dr. Hudson Tercio Pinheiro 

Academic Editor

PLOS ONE